# Interferon-γ signaling synergizes with LRRK2 in neurons and microglia derived from human induced pluripotent stem cells

Vasiliki Panagiotakopoulou[1,2,6], Dina Ivanyuk[1,2,6], Silvia De Cicco[1,2,6], Wadood Haq[3], Aleksandra Arsić[4], Cong Yu[1,2], Daria Messelodi[1,2], Marvin Oldrati[1,2], David C. Schöndorf[1,2], Maria-Jose Perez[1,2], Ruggiero Pio Cassatella[1,2], Meike Jakobi[5], Nicole Schneiderhan-Marra[5], Thomas Gasser[1,2], Ivana Nikić-Spiegel[4] & Michela Deleidi[1,2 ✉]

Parkinson's disease-associated kinase LRRK2 has been linked to IFN type II (IFN-γ) response in infections and to dopaminergic neuronal loss. However, whether and how LRRK2 synergizes with IFN-γ remains unclear. In this study, we employed dopaminergic neurons and microglia differentiated from patient-derived induced pluripotent stem cells carrying *LRRK2* G2019S, the most common Parkinson's disease-associated mutation. We show that IFN-γ enhances the *LRRK2* G2019S-dependent negative regulation of AKT phosphorylation and NFAT activation, thereby increasing neuronal vulnerability to immune challenge. Mechanistically, *LRRK2* G2019S suppresses NFAT translocation via calcium signaling and possibly through microtubule reorganization. In microglia, LRRK2 modulates cytokine production and the glycolytic switch in response to IFN-γ in an NFAT-independent manner. Activated *LRRK2* G2019S microglia cause neurite shortening, indicating that LRRK2-driven immunological changes can be neurotoxic. We propose that synergistic LRRK2/IFN-γ activation serves as a potential link between inflammation and neurodegeneration in Parkinson's disease.

[1] German Center for Neurodegenerative Diseases (DZNE), Tübingen 72076, Germany. [2] Department of Neurodegenerative Diseases, Hertie-Institute for Clinical Brain Research, University of Tübingen, Tübingen 72076, Germany. [3] Centre for Ophthalmology, Institute for Ophthalmic Research University of Tübingen, University of Tübingen, Tübingen 72076, Germany. [4] Werner Reichardt Centre for Integrative Neuroscience, University of Tübingen, Tübingen 72076, Germany. [5] NMI Natural and Medical Sciences Institute at the University of Tübingen, 72770 Reutlingen, Germany. [6]These authors contributed equally: Vasiliki Panagiotakopoulou, Dina Ivanyuk, Silvia De Cicco. ✉email: michela.deleidi@dzne.de

Emerging data suggest that immune dysregulation contributes to not only the progression, but also the onset of neurodegenerative diseases such as Parkinson's disease (PD)[1]. In this respect, human genetics and functional genomics studies indicate that interferon (IFN)-mediated signaling pathways, including those of IFN type I and type II, play a role in brain aging and human neurodegenerative diseases[2,3]. With regard to PD, there is evidence for a link between expression network signatures of disease loci and IFN type II—also known as IFN-γ— signaling[4]. Recently, IFN-γ signaling has also been linked to the selective vulnerability of dopaminergic (DA) neurons[5].

IFN-γ is a cytokine primarily produced by T-lymphocytes and natural killer cells as part of the normal immune response against pathogens. IFN-γ plays a key role in both innate and adaptive immune responses and contributes to macrophage activation, upregulation of major histocompatibility complex (MHC) proteins, and modulation of the T-helper cell response. Besides its role in immune responses against pathogens, IFN-γ is also fundamental to normal brain physiology[6]. The high incidence of postencephalitic parkinsonism following the influenza pandemic near the end of World War I had already suggested vulnerability of DA neurons to pathogen-driven immune responses. Indeed, pathological studies revealed the presence of inflammatory encephalitis in the midbrain and basal ganglia, without central nervous system viral invasion[7].

Supporting a possible role of IFN-γ in pathological processes, increased IFN-γ levels in the serum, and increased IFN-γ production by peripheral CD4+ T cells relative to healthy controls have been detected in PD patients[8,9]. Moreover, higher levels of IFN-γ, with a significant co-expression of α-synuclein, have been found in the substantia nigra of PD patients[10,11]. Experimental models also corroborate this connection: IFN-γ has been linked to progressive DA neurodegeneration in rodents[8,12]. More recent studies further support the interaction between PD-related genes and IFN-γ induced DA neuronal loss[13]. For instance, Kozina et al. showed that pathogenic mutations in the human leucine-rich repeat kinase 2 (LRRK2) gene, the most common genetic cause of familial PD[14], synergizes with lipopolysaccharide (LPS)-induced inflammation to potentiate DA neurodegeneration through IFN-γ-mediated immune responses[13]. Interestingly, LRRK2 is an IFN-γ target gene[15,16], and its expression is upregulated in immune cells following IFN-γ stimulation and exposure to pathogens[15,17–20]. Genetic polymorphisms in LRRK2 have also been associated with Crohn's disease and leprosy[21,22], suggesting overlapping pathogenetic mechanisms among chronic inflammatory diseases, infections, and PD. PD patients carrying LRRK2 mutations (LRRK2-PD) manifest clinical phenotypes similar to those of idiopathic PD, displaying a strong age-dependent development of PD symptoms.

Notably, PD patients carrying LRRK2 G2019S—the most common PD-associated mutation—have distinct peripheral inflammatory profiles[23]. Despite the correlation between IFN-γ signaling and PD, the mechanisms through which IFN-γ contributes to neurodegeneration and interacts with PD genes, such as LRRK2, remain unknown.

In this study, we investigate mechanisms of IFN-γ-mediated neurotoxicity and explore the role of the pathogenic PD-associated LRRK2 mutation G2019S in IFN-γ signaling modulation. We show that G2019S mutation sensitizes neurons to IFN-γ signaling and we provide evidence that LRRK2 regulates microglial response to IFN-γ by modulating immune metabolic reprogramming. These findings have potential therapeutic implications as compounds targeting LRRK2 are currently being tested in clinical trials for PD.

## Results

**IFN-γ induces LRRK2 expression in human neurons.** To examine the impact of IFN-γ signaling on human neurons, we employed induced pluripotent stem cell (iPSC)-derived neural precursor cells (NPCs). Control NPCs were differentiated into neuronal cultures, enriched into DA neurons, and treated with IFN-γ (100, 200, or 400 IU/mL for 24 h). Since previous studies showed that IFN-γ induces LRRK2 expression in immune cells[15,17–19], we examined LRRK2 mRNA and protein levels in untreated and IFN-γ-treated neuronal cultures. IFN-γ stimulation significantly increased LRRK2 mRNA and protein levels in control neurons (Fig. 1a, b). To evaluate whether PD-associated LRRK2 missense mutations influence the IFN-γ-driven induction of LRRK2 expression, we used NPC lines derived from patients carrying the LRRK2 G2019S mutation and compared them to corresponding isogenic controls[24,25]. As previously reported[25], isogenic NPCs were differentiated into DA neurons without significant differences among patients and controls (Supplementary Fig. 1A, B). LRRK2 protein expression was similar among different lines (Supplementary Fig. 1C). IFN-γ treatment also upregulated LRRK2 protein levels in LRRK2 G2019S PD patient-derived DA neurons (Fig. 1b). Based on these results, a dose of 200 IU/mL IFN-γ was deemed optimal and used in subsequent experiments. IFN-γ also increased LRRK2 levels in iPSC-derived neuronal cultures enriched in cortical neurons, suggesting that this effect is not specific to DA neurons (Supplementary Fig. 1D). IFN-γ treatment did not affect neuronal cell viability, as assessed by LDH assay (Supplementary Fig. 1E). To examine the specificity of the effects observed with IFN-γ, control DA neurons were treated with IFN-γ, LPS, or IL-1β for 24 h, and LRRK2 mRNA levels were measured using qRT-PCR; only IFN-γ was able to induce LRRK2 expression in human neurons (Supplementary Fig. 1F).

**LRRK2 G2019S alters IFN-γ signaling in human neurons.** To characterize the IFN-γ response in human neurons, healthy control NPC lines were differentiated into DA neurons and treated with 200 IU/mL IFN-γ for 24 h. IFN-γ-regulated genes, selected according to the Interferome v2.0 gene expression database[26] were characterized by qRT-PCR in vehicle- and IFN-γ-treated neurons. Interferon-gamma receptor 1 (IFNGR1), proto-oncogene, non-receptor tyrosine kinase (SRC), and mitogen-activated protein kinase 14 (MAPK14) were significantly downregulated (Fig. 1c), whereas Janus kinase 2 (JAK2), eukaryotic translation initiation factor 2-alpha kinase 2 (EIF2AK2), AKT serine/threonine kinase 3 (AKT3), signal transducer and activator of transcription 1 (STAT1), suppressor of cytokine signaling 1 (SOCS1), intercellular adhesion molecule-1 (ICAM1), and interferon regulatory factor 1 (IRF1) were significantly upregulated in IFN-γ treated neurons relative to vehicle-treated neurons (Fig. 1c). To examine whether LRRK2 G2019S influences the neuronal response to IFN-γ, we measured the expression of the most strongly upregulated (i.e., >5-fold) genes upon IFN-γ treatment in LRRK2 G2019S mutant NPC-derived neurons and their corresponding isogenic controls. Among the upregulated genes analyzed, significantly higher mRNA levels of AKT3 were found in LRRK2 G2019S mutant neurons upon IFN-γ stimulation than were found in controls (Fig. 1d). Similar results were confirmed in two additional pairs of LRRK2 G2019S and isogenic gene-corrected control lines (Fig. 1e). At the protein level, we detected a modest, non-significant AKT3 increase in IFN-γ-treated cells (Fig. 1f). AKT3 phosphorylation was assessed by Luminex multiplex assay. Basal phospho-AKT3 levels were lower in LRRK2 G2019S neurons than in controls (Fig. 1g). IFN-γ treatment significantly reduced AKT3 phosphorylation in both

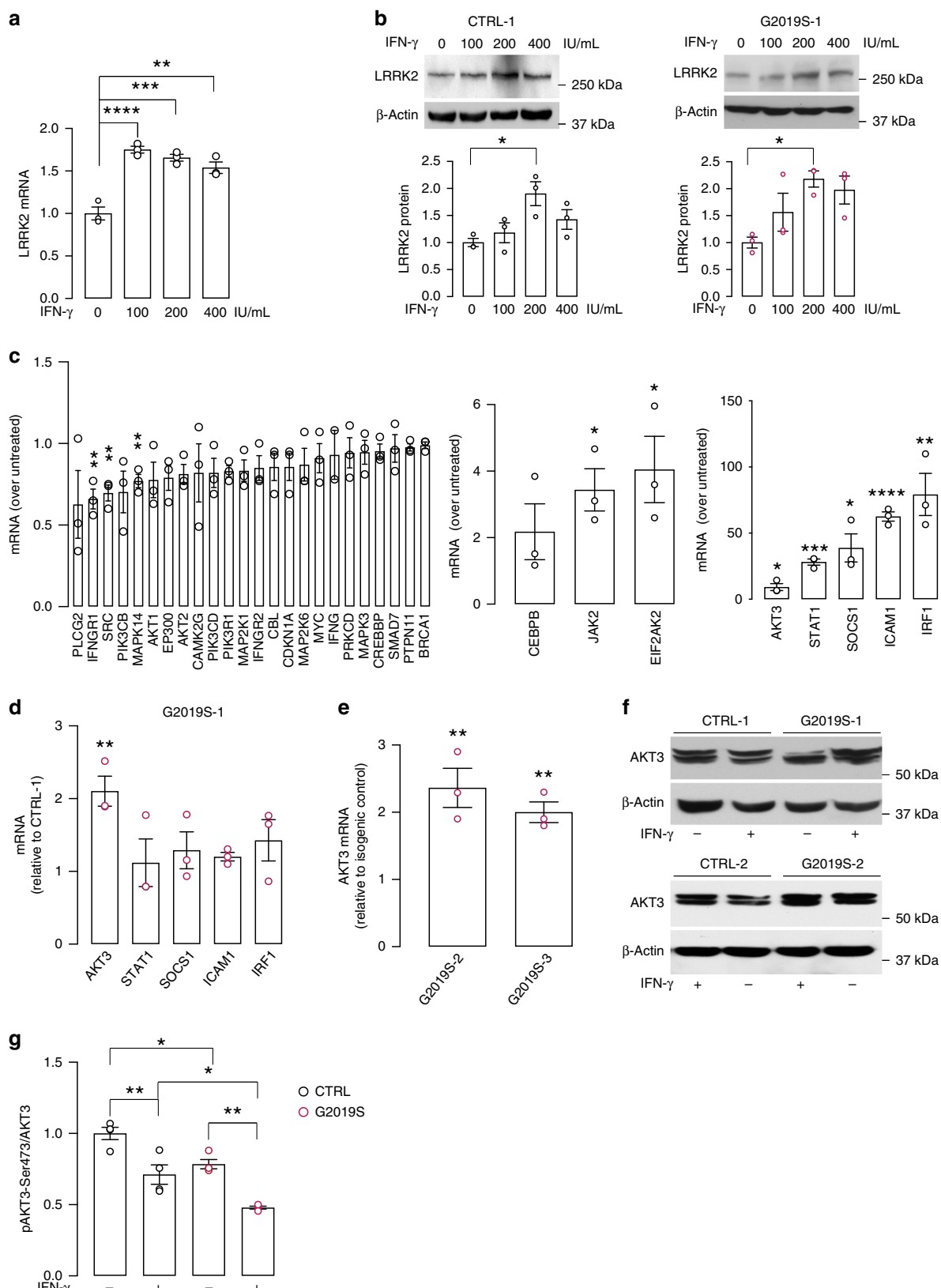

LRRK2 G2019S and control neurons, with the strongest effect observed in LRRK2 G2019S neurons (38.9% reduction, compared to 28.8% in control neurons) (Fig. 1g). To confirm this result, and to determine whether all AKT isoforms are similarly regulated by LRRK2 and IFN-γ, we performed Western blotting using a pan-AKT antibody. In line with our phospho-AKT3 data,

phospho-AKT (Ser473) levels were lower in LRRK2 G2019S neurons than in control neurons, and IFN-γ treatment reduced AKT phosphorylation in both LRRK2 G2019S neurons and controls, with a more pronounced effect in LRRK2 G2019S neurons (33.2% reduction, compared to 23.7% in control neurons) (Supplementary Fig. 1G, H).

**Fig. 1 IFN-γ induces LRRK2 expression in human iPSC-derived neurons and decreases AKT phosphorylation.** IFN-γ treatment induces an increase of LRRK2 levels in human iPSC-derived neurons. **a** *LRRK2* mRNA levels in control iPSC-derived neurons treated with IFN-γ for 24 h at the indicated concentrations. *LRRK2* mRNA levels, measured by qRT-PCR, are expressed as fold changes relative to untreated (mean ± SEM, one-way ANOVA, Bonferroni post hoc, ****$P = 0.0001$, ***$P = 0.0003$, **$P = 0.0011$, $n = 3$ independent experiments). **b** Representative Western blots and corresponding quantification of LRRK2 expression in control (left) and isogenic *LRRK2* G2019S (right panel) iPSC-derived neurons treated with IFN-γ at the indicated concentrations (mean ± SEM, one-way ANOVA, Bonferroni post hoc, *$P < 0.05$; $n = 3$ independent experiments). **c** IFN-γ transcriptional signature genes in control iPSC-derived neurons treated with 200 IU/mL IFN-γ for 24 h measured by qRT-PCR and expressed as fold changes relative to untreated controls. Downregulated (left panel) and moderately (fold change <5, middle panel) or strongly (fold change >5, right panel) upregulated genes are shown (mean ± SEM, two-tailed t-test treated vs untreated, left panel: **$P = 0.0075, 0.0075, 0.0052$ in sequence, middle panel: *$P = 0.0186, 0.0379$ in sequence, right panel: ****$P = 0.0001$, ***$P = 0.0003$, **$P = 0.0080$, *$P = 0.0356, 0.0238$ in sequence; $n = 3$ independent experiments). **d** mRNA levels of AKT3, STAT1, SOCS1, ICAM1, and IRF1 in *LRRK2* G2019S (G2019S-1) neurons treated with 200 IU/mL IFN-γ, expressed as fold changes relative to corresponding IFN-γ treated isogenic controls (CTRL-1) (mean ± SEM, two-tailed *t*-test, **$P = 0.0060$; $n = 3$ independent experiments). **e** mRNA levels of AKT3 in *LRRK2* G2019S neurons (G2019S-2, G2019S-3) treated with 200 IU/mL IFN-γ, expressed as fold changes relative to corresponding IFN-γ-treated isogenic controls (CTRL-2, CTRL-3) (mean ± SEM, two-tailed *t*-test, **$P = 0.0088, 0.0033$ in sequence; $n = 3$ independent experiments). **f** Representative Western blot of AKT3 in *LRRK2* G2019S neurons (G2019S-1, G2019S-2) and corresponding isogenic controls (CTRL-1, CTRL-2). Treatment with 200 IU/mL IFN-γ is indicated ($n = 3$ independent experiments). **g** Quantification of phospho-AKT3/AKT3 ratio in *LRRK2* G2019S neurons and isogenic controls, untreated and upon IFN-γ treatment, as assessed by Luminex multiplex analysis (mean ± SEM, two-way ANOVA, Bonferroni post hoc, **$P = 0.0033, 0.0021$ in sequence, *$P = 0.0270, 0.0165$ in sequence; $n = 4$ independent experiments).

**LRRK2 G2019S reduces nuclear NFAT3 in human neurons.** Since AKT promotes the nuclear retention and activity of nuclear factor of activated T-cells (NFAT)[27,28], we investigated the possible impact of reduced AKT phosphorylation (i.e., reduced AKT activity) on NFAT shuttling. NFAT family members are regulators of $Ca^{2+}$-dependent gene transcription in a variety of cell types, including immune cells and neurons[29]. In the brain, NFAT transcription complexes play important roles in neuronal growth and brain circuit formation[29]. We first examined the expression levels of the $Ca^{2+}$-dependent NFAT isoforms (NFAT1, NFAT2, NFAT3, and NFAT4) to control human iPSC-derived neurons by qRT-PCR (Supplementary Fig. 1I). Given that NFAT3 was most highly expressed in neurons, we focused on this isoform. We then assessed NFAT3 levels in nuclear and cytosolic fractions from *LRRK2* G2019S neurons and isogenic controls, with and without IFN-γ treatment (Fig. 2a). Notably, we observed a significant reduction in nuclear NFAT3 levels in *LRRK2* G2019S neurons compared to controls (Fig. 2a), suggesting impaired basal translocation. Furthermore, IFN-γ treatment significantly reduced nuclear NFAT3 levels in both *LRRK2* G2019S and controls (Fig. 2a). To investigate the activation of the NFAT pathway in DA neurons, we treated iPSC-derived neurons with ionomycin and performed nuclear/cytosolic fractionation. Ionomycin treatment induced the nuclear translocation of NFAT3 in both control neurons and *LRRK2* G2019S iPSC-derived neurons, an effect that was significantly reduced following IFN-γ treatment (Fig. 2b, Supplementary Fig. 1J). We then analyzed Western blot results for total NFAT3 levels in *LRRK2* G2019S and control neurons with and without IFN-γ treatment. No changes were observed in total NFAT3 levels at basal conditions, but a significant reduction was observed upon treatment with IFN-γ in both control and *LRRK2* G2019S neurons (Fig. 2c). Given that IFN-γ activates the proteasome[30], we examined whether the reduction of total NFAT3 levels observed in IFN-γ-treated neurons could be mediated by proteasome activation. To this end, control neurons were treated with IFN-γ with or without proteasome inhibition by MG132, and NFAT3 levels were assessed by Western blotting. Interestingly, proteasome inhibition restored NFAT3 levels in IFN-γ treated neurons (Fig. 2d). To confirm that LRRK2 acts downstream of IFN-γ, we generated *LRRK2* knockout (KO) iPSCs (Supplementary Fig. 2A). *LRRK2* KO iPSCs and isogenic controls were differentiated into DA neurons (Supplementary Fig. 2B, C) and treated with IFN-γ or vehicle. In line with previous reports[31], we observed increased nuclear NFAT3 levels in *LRRK2* KO neurons compared to isogenic controls, both at basal conditions

and after treatment with ionomycin (Supplementary Fig. 2D), whereas IFN-γ had no significant impact on nuclear NFAT3 levels in *LRRK2* KO neurons (Fig. 2e).

**LRRK2 alters the ER-related neuronal calcium response to IFN-γ.** We next sought to explore mechanisms that could link the defects of NFAT translocation to the LRRK2/IFN-γ axis. One of the key pathways initiating NFAT shuttling is calcium mobilization through the activation of store-operated $Ca^{2+}$ entry (SOCE)[32]. Hence, we examined *LRRK2* G2019S-related changes in $Ca^{2+}$ homeostasis by Fura-2 live-cell $Ca^{2+}$ imaging in *LRRK2* G2019S neurons and corresponding isogenic controls. Upon inhibition of sarco(endo)plasmic reticulum $Ca^{2+}$-ATPase (SERCA) with thapsigargin (TPH), we observed significantly reduced $Ca^{2+}$ release from the endoplasmic reticulum (ER) of mutant cell lines, indicating the impaired capacity of the ER to store $Ca^{2+}$ in *LRRK2* G2019S mutant neurons (Fig. 3a, b). To examine the effect of chronic IFN-γ treatment on neuronal $Ca^{2+}$ dynamics, *LRRK2* G2019S mutant and control neurons were treated with 200 IU/mL IFN-γ for 24 h and the ER $Ca^{2+}$-release in response to TPH was assessed by live-cell $Ca^{2+}$ imaging. We observed a significant reduction of the ER $Ca^{2+}$-release induced by TPH upon IFN-γ treatment in control cell lines (Fig. 3c, d). Conversely, no significant difference was found in IFN-γ-treated *LRRK2* G2019S mutant neurons, supporting a pre-existing depletion of ER $Ca^{2+}$-stores (Fig. 3c, d).

**LRRK2 regulates microtubule network organization in human neurons.** The integrity of the microtubule network is necessary for NFAT translocation[33,34]. Furthermore, LRRK2 interacts with and regulates several cytoskeletal components in a variety of cell types[35]. However, LRRK2-mediated morphological changes of the microtubule network have never been investigated.

To detect LRRK2-related changes in microtubule morphology at nanoscale resolution, we stained control and isogenic *LRRK2* G2019S iPSC-derived neurons for βIII-tubulin with an Alexa Fluor 647-conjugated secondary antibody, followed by super-resolution imaging with direct stochastic optical reconstruction microscopy (dSTORM) (Fig. 4a). The quantitative analysis of the cell soma microtubule network was performed using a recent computational tool designed by Zhang et al. for automated extraction of microtubule filaments from single-molecule-localization–based superresolution microscopy images (SIFNE)[36]. Analysis of the dSTORM images revealed that the cell soma area and total number

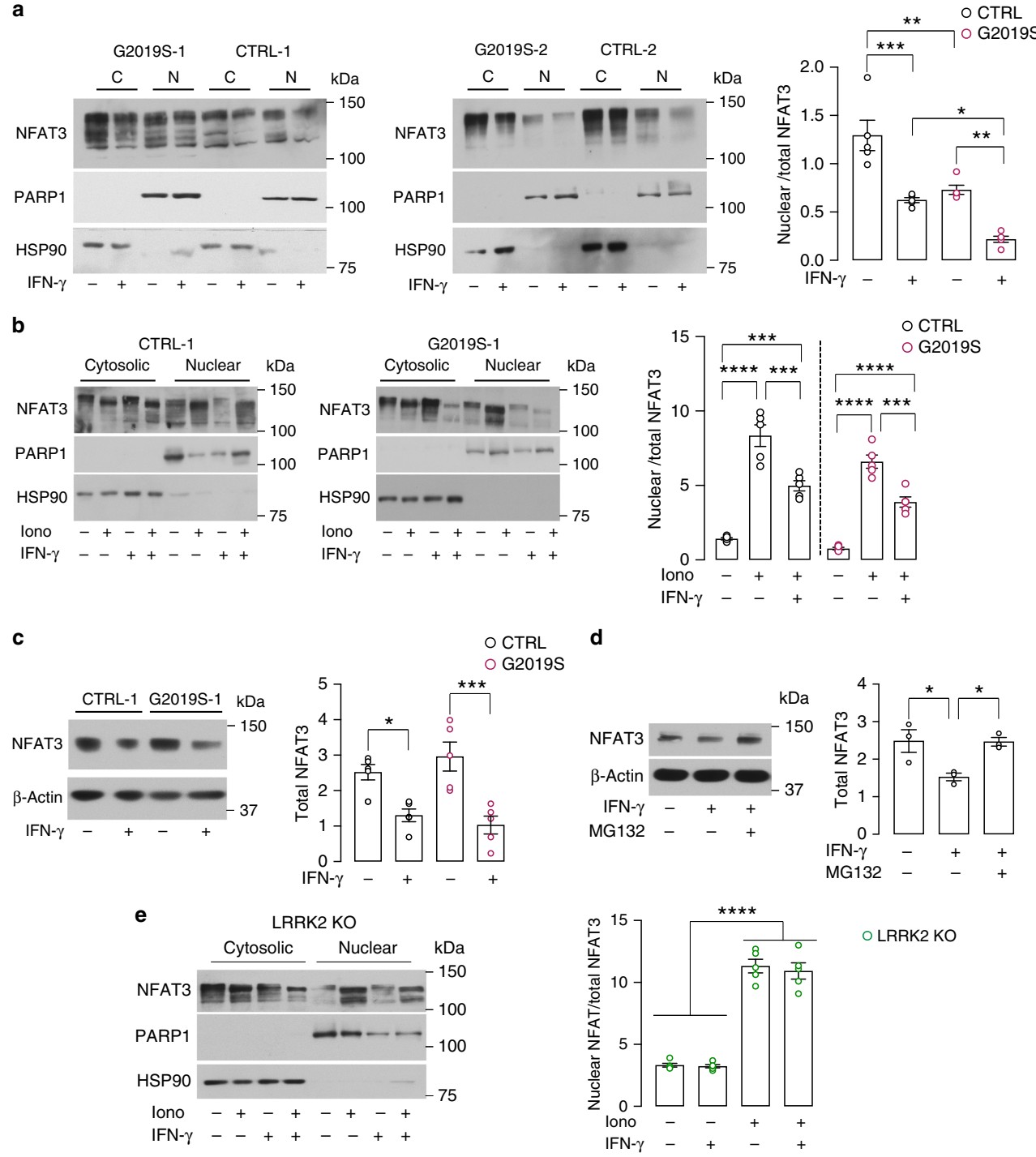

of filaments did not vary between cell lines (Fig. 4b, c), but the average filament length was found to be shorter in *LRRK2* G2019S neurons than in controls (Fig. 4d). Moreover, while the total number of junctions was unaffected (Fig. 4e), a higher distribution of junctions, higher curvatures, and different curvature sizes at the cell periphery were found in *LRRK2* G2019S neurons relative to controls, suggesting a different network complexity (Fig. 4f–h).

To strengthen the link between the microtubule network organization and NFAT shuttling, we employed HEK reporter cell lines expressing luciferase under a promoter with NFAT-response elements and assessed the effect of microtubule-stabilizing and destabilizing agents. Treatment with the microtubule polymerization inhibitor colchicine significantly reduced NFAT

activity, whereas stabilizing microtubules with paclitaxel (PTX) significantly increased NFAT activation (Supplementary Fig. 3A, B). Next, we transfected HEK NFAT reporter cell lines with *LRRK2* wild-type (wt) or *LRRK2* G2019S constructs (Supplementary Fig. 3C) and measured NFAT transcriptional activity in response to phorbol 12-myristate 13-acetate (PMA) and ionomycin. We found that overexpression of wt *LRRK2* inhibits NFAT-dependent transcriptional activity (Supplementary Fig. 3D). Moreover, NFAT activity was decreased to a greater extent in cells overexpressing *LRRK2* G2019S (Supplementary Fig. 3D). PTX treatment effectively rescued the inhibitory effect of wt *LRRK2* and *LRRK2* G2019S overexpression on NFAT activation, suggesting that microtubule stabilization promotes NFAT nuclear shuttling

**Fig. 2 LRRK2 inhibits nuclear NFAT3 shuttling in human neurons. a** Representative Western blots of NFAT3 in nuclear (N) and cytosolic (C) fractions of *LRRK2* G2019S neurons and isogenic controls (left and middle panel). Treatment with 200 IU/mL IFN-γ is indicated. On the right, quantification of nuclear to total (N and C fraction) NFAT3 in *LRRK2* G2019S iPSC-derived neurons and isogenic controls (mean ± SEM, two-way ANOVA, Bonferroni post hoc, ***$P = 0.0003$, **$P = 0.0015$, $0.0035$ in sequence, *$P = 0.0226$; $n = 5$ independent experiments). **b** Representative Western blots of NFAT3 in N and C fractions of control (left panel) and isogenic *LRRK2* G2019S neurons (middle panel). Treatments with 1 μM ionomycin for 30 min and 200 IU/mL IFN-γ for 24 h are indicated. On the right, quantification of nuclear to total (N and C fraction) NFAT3 in *LRRK2* G2019S iPSC-derived neurons and isogenic controls (mean ± SEM, one-way ANOVA, Bonferroni post hoc, ****$P < 0.0001$, ***$P = 0.0005$, $0.0008$, $0.0002$ in sequence; $n = 5$ independent experiments). **c** Representative Western blot (left) and quantification (right) of NFAT3 from whole-cell extracts in *LRRK2* G2019S neurons and isogenic controls. Treatments with 200 IU/mL IFN-γ are indicated (mean ± SEM, two-way ANOVA, Bonferroni post hoc, ***$P = 0.0009$, *$P = 0.0421$; $n = 5$ independent experiments). **d** Representative Western blot (left) and quantification (right) of NFAT3 from whole-cell extracts in control iPSC-derived neurons upon treatment with 200 IU/mL IFN-γ for 24 h, with and without treatment with a proteasome inhibitor (MG132, 20 nM for 24 h) (mean ± SEM, one-way ANOVA, Bonferroni post hoc, *$P = 0.0399$, $0.0437$ in sequence; $n = 3$ independent experiments). **e** Representative Western blots of NFAT3 in nuclear and cytosolic fractions of *LRRK2* KO neurons. Treatments with 1 μM ionomycin for 30 min and 200 IU/mL IFN-γ for 24 h are indicated. On the right, quantification of nuclear to total (N and C fraction) NFAT3 (mean ± SEM, one-way ANOVA, Bonferroni post hoc, ****$P < 0.0001$; $n = 5$ independent experiments).

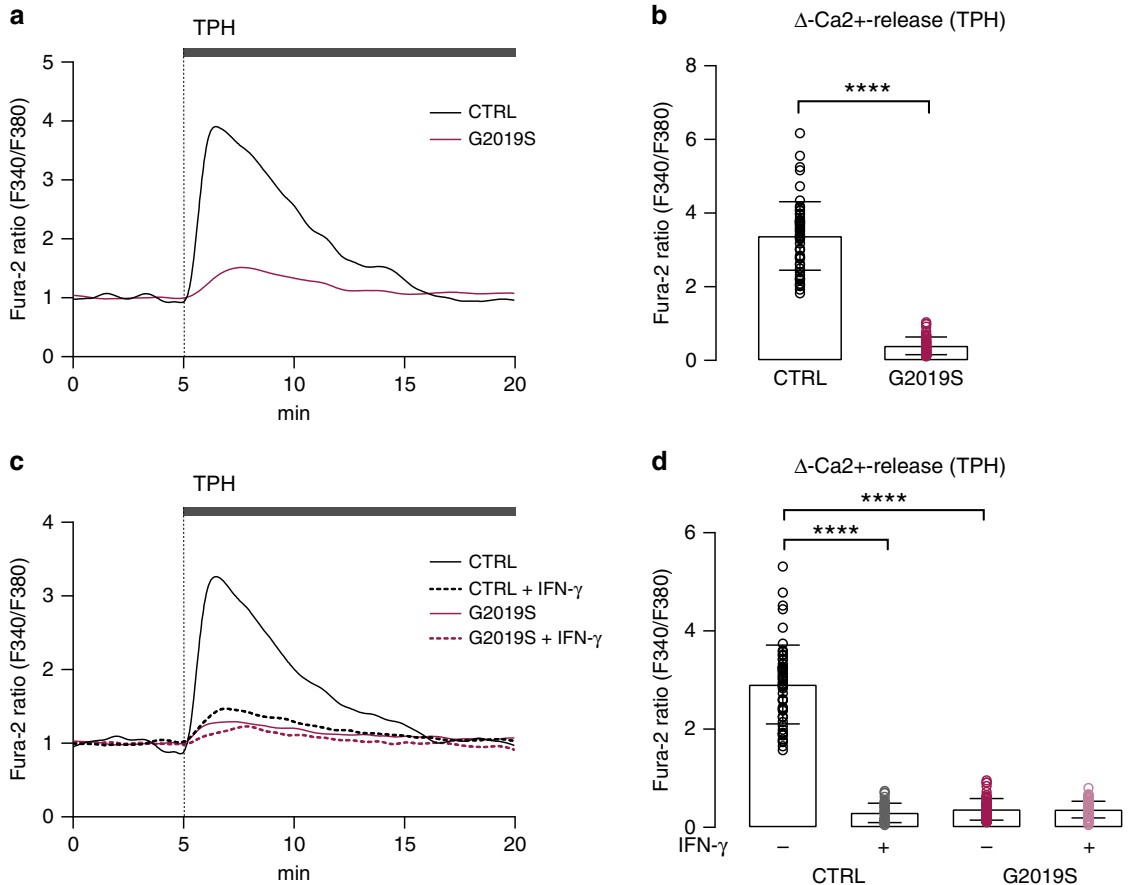

**Fig. 3 LRRK2 G2019S alters Ca$^{2+}$-storage and Ca$^{2+}$ response to IFN-γ.** ER Ca$^{2+}$ release was measured after SERCA inhibition with thapsigargin (TPH, 500 nM) in human iPSC-derived neurons (G2019S and isogenic controls) using fura-2 AM ratiometric Ca$^{2+}$ imaging. **a** Individual Ca$^{2+}$ signaling traces of representative isogenic *LRRK2* G2019S and control neurons are shown. The change in fluorescence was normalized to the corresponding baseline fluorescence prior to TPH stimulation. **b** Quantification of Ca$^{2+}$ peak amplitude (Δ-Ca$^{2+}$ release) is shown (mean ± SEM, two-tailed t-test, ****$P < 0.0001$; $n = 56$ control and 59 G2019S individual cells examined over three independent experiments; non-normalized data are provided in the Data Source File). **c** Individual Ca$^{2+}$ signaling traces of representative isogenic *LRRK2* G2019S and control neurons pretreated with 200 IU/mL IFN-γ for 24 h or left untreated. **d** Quantification of TPH-mediated Δ-Ca$^{2+}$ release is shown (mean ± SEM, two-way ANOVA, Bonferroni post hoc, ****$P < 0.0001$; $n = 56$ control, 42 IFN-γ treated control, 59 G2019S and 59 IFN-γ treated G2019S individual cells examined over three independent experiments).

(Supplementary Fig. 3D). Interestingly, the rescuing effect of PTX was more pronounced in the *LRRK2* G2019S cells. Indeed, the ratio of luciferase expression in PTX-treated vs untreated ionomycin/PMA-stimulated cells was higher in G2019S cells than in to both non-transfected and wt *LRRK2*-overexpressing cells (1.71 vs 1.30 and 1.45, respectively).

These data suggest that LRRK2-associated changes in the microtubule network are involved in NFAT shuttling. To validate these findings in a more disease-relevant context, isogenic control and *LRRK2* G2019S neurons were treated with PTX and immunostained for phospho-NFAT3, the cytosolic, inactive form of NFAT3. Basal levels of phospho-NFAT3 were significantly

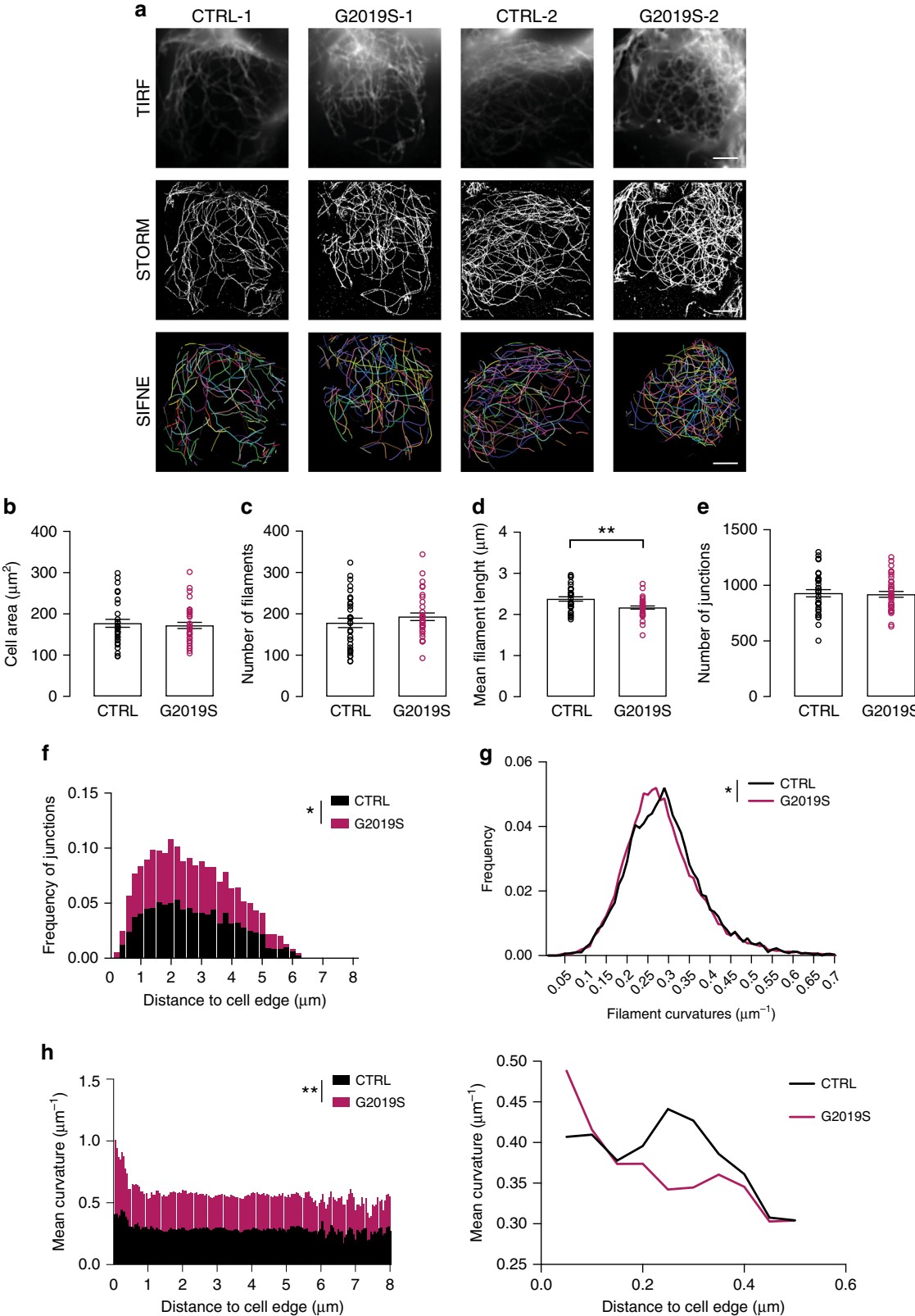

higher in *LRRK2* G2019S neurons than in controls, further corroborating the inhibitory effect of the PD-associated *LRRK2* mutation on NFAT shuttling (Supplementary Fig. 3E, F). Microtubule stabilization with PTX significantly decreased the levels of phospho-NFAT3, indicating increased NFAT3 nuclear translocation (Supplementary Fig. 3E, F). These findings support

the link between LRRK2-mediated changes in the microtubule network and impairment of NFAT shuttling.

**IFN-γ impairs neurite outgrowth in human iPSC-derived neurons**. We next inquired as to whether the reduced NFAT3 nuclear localization observed in *LRRK2* G2019S neurons has

**Fig. 4 Superresolution imaging of LRRK2-dependent microtubule network. a** Representative images of iPSC-derived neurons carrying the *LRRK2* G2019S mutation and corresponding isogenic controls (3 independent experiments were performed and 5-7 cells were analyzed per condition, per experiment). Upper panel: total internal reflection fluorescence (TIRF) images of cell soma microtubules stained for β-III tubulin and probed with Alexa Fluor 647-conjugated secondary antibody; middle panel: superresolution dSTORM images of microtubule networks; lower panel: extraction of filaments network, composite filament network obtained from automated extraction by SIFNE. Scale bars, 5 μm. **b–h** Quantitative analysis of the microtubule network. **b** cell soma area (mean ± SEM, two-tailed *t*-test, $n = 33$ control and 36 G2019S individual cells examined over three independent experiments); **c** number of filaments per cell (mean ± SEM, two-tailed *t*-test, $n = 33$ control and 36 G2019S individual cells examined over three independent experiments); **d** mean filament length (mean ± SEM, two-tailed *t*-test, $**P = 0.0023$, $n = 31$ control and 35 G2019S individual cells examined over three independent experiments); **e** number of junctions (mean ± SEM, two-tailed *t*-test, $n = 33$ control and 35 G2019S individual cells examined over three independent experiments); **f** normalized distributions of junction density as a function of distance from cell edge (mean ± SEM, multiple *t*-test, $*P = 0.0005$, $n = 33$ control and 36 G2019S individual cells examined over three independent experiments); **g** normalized filament curvature distribution (mean ± SEM, multiple *t*-test, $*P = 0.0003$, $n = 33$ control and 36 G2019S individual cells examined over three independent experiments); **h** distribution of mean curvature as a function of distance from cell soma edges Kolmogorov–Smirnov test, $**P = 0.0015$; $n = 33$ control and 36 G2019S individual cells examined over three independent experiments); on the right, a more detailed representation of the values in close proximity to the cell edge (0–0.5 μm).

functional consequences relevant to human disease. Given that NFAT signaling is essential for axonal outgrowth in response to neurotrophins and netrins[37], we examined whether the reported LRRK2-mediated defects in neurite outgrowth[25,38,39] could be also linked to NFAT signaling. To this end, *LRRK2* G2019S neurons and corresponding isogenic controls were treated with either IFN-γ or MCV1, a potent and specific inhibitor of calcineurin-mediated NFAT activation, and neurite length was quantified by immunostaining for βIII-tubulin. NFAT inhibition, as well as IFN-γ treatment, led to a significant decrease in neurite length in both control and *LRRK2* G2019S iPSC-derived neurons (Fig. 5a, b). Notably, treatment with different LRRK2 kinase inhibitors (IN-1, GSK2578215A, and Mli-2) rescued these neurite length defects following IFN-γ treatment (Supplementary Fig. 4A, B). To confirm our hypothesis that the reduction of neurite length was mediated by decreased NFAT activity, we performed a rescue experiment with neuregulin 1 (NRG1), which promotes NFAT activity[40]. Control and *LRRK2* G2019S neurons were treated with MCV1 or IFN-γ with or without NRG1. Our results revealed that NRG1 rescues neurite length defects in both MCV1 and IFN-γ-treated control neurons (Fig. 5c, d). NRG1 treatment also partially protected neurite length defects in *LRRK2* G2019S neurons (Fig. 5c, d). These effects on neurite length were confirmed in DA neurons by immunostaining for tyrosine hydroxylase (TH) (Supplementary Fig. 4C, D, Supplementary Fig. 5A, B). To examine whether the restorative effects of NRG1 were mediated by its effect on calcium dynamics, we assessed ER Ca²⁺ release in response to TPH in control and *LRRK2* G2019S neurons treated with or without NRG1. NRG1 treatment was able to rescue the impaired Ca²⁺-storage capacity of the ER (Supplementary Fig. 5C, D).

**LRRK2 regulates NFAT shuttling in human macrophages and microglia.** NFAT regulates responses within the innate and adaptive arms of the immune system[41]. Since NFAT1 was the main NFAT isoform expressed in human microglia (Supplementary Fig. 6A), this isoform was analyzed in myeloid cells. We confirmed that IFN-γ increases LRRK2 expression in mononuclear phagocytes by treating human macrophages differentiated from the monocytic cell line THP-1 as well as human iPSC microglia with 200 IU/mL IFN-γ for 72 h. Similar to that observed in neurons, LRRK2 expression was increased after IFN-γ stimulation (Supplementary Fig. 6B). To investigate the role of LRRK2 in NFAT1 shuttling in human macrophages, we knocked down *LRRK2* (*LRRK2* KD) via lentiviral-mediated shRNA delivery in THP-1 macrophages (Supplementary Fig. 6C, D). *LRRK2* KD THP-1 macrophages displayed increased NFAT1 nuclear shuttling upon stimulation with ionomycin (Supplementary Fig. 6E, F). Control *LRRK2* KO

iPSCs and *LRRK2* G2019S iPSCs differentiated into microglia without significant differences (Fig. 6a–c, Supplementary Fig. 7A). Interestingly, microglia motility was increased in *LRRK2* G2019S compared to control cells (Fig. 6d). However, IFN-γ significantly decreased motility of G2019S, but not control, microglia (Fig. 6d). Moreover, *LRRK2* G2019S microglia displayed significantly higher phagocytic capacity compared to control cells, suggesting a more active phenotype (Fig. 6e). Immunofluorescent analysis of NFAT1 subcellular localization revealed increased levels of nuclear NFAT in *LRRK2* KO microglia treated with ionomycin (Supplementary Fig. 7A, B). Conversely, nuclear NFAT1 levels were significantly lower in *LRRK2* G2019S microglia (Supplementary Fig. 7A, B). Given that ionomycin is a weak activator and stimulator of cytokine release in microglia, we tested the impact of LPS stimulation on NFAT nuclear shuttling, based on previous reports[42]. Nuclear NFAT1 was significantly increased in *LRRK2* KO microglia compared to controls upon LPS treatment, whereas a significant reduction was observed in LPS-stimulated *LRRK2* G2019S cells compared to isogenic controls (Supplementary Fig. 7A, C). To assess whether the LRRK2-associated changes in the microtubule network that were observed in neurons could also be linked to defects in nuclear translocation of other transcription factors relevant to microglial function, we evaluated NF-κB p65 localization in LPS- and IFN-γ-treated microglia. Interestingly, we detected impairment in NF-κB p65 nuclear translocation in *LRRK2* G2019S microglia primed with LPS or IFN-γ compared to isogenic controls (Supplementary Fig. 8A, B).

We next examined the functional consequences of NFAT activity in human microglia. Control, G2019S, and *LRRK2* KO iPSC-derived microglia were treated with LPS, and cytokine production was assessed by multiplex ELISA (Fig. 6f, Supplementary Fig. 9A). Among the analyzed cytokines, IL-6, TNF-α, and IL-8 were significantly decreased in G2019S microglia compared to isogenic controls, whereas IL-10, IL-1β, IL-12p70, VEGF, and MIP-1β were significantly increased, upon LPS stimulation (Fig. 6f, Supplementary Fig. 9A). On the other hand, we detected a significant upregulation of IL-6 and MIP-1β and a significant downregulation of IL-1β and IL-10 secretion in LPS-treated *LRRK2* KO microglia compared to isogenic controls (Fig. 6f). To strengthen the link between LRRK2 and microglia activation, we assessed the impact of LRRK2 kinase inhibition on the levels of various cytokines upon LPS treatment in isogenic control and G2019S microglia. In control microglia, LRRK2 kinase inhibition significantly reduced IL-6, TNF-α, IL-8, IL1-β, IL-10, MIP-1β, and MCP-1 levels and significantly increased VEGF and IL-12p70 levels. On the other hand, only IL-8, IL-12p70, MIP-1β, and IL-1RA levels were reduced

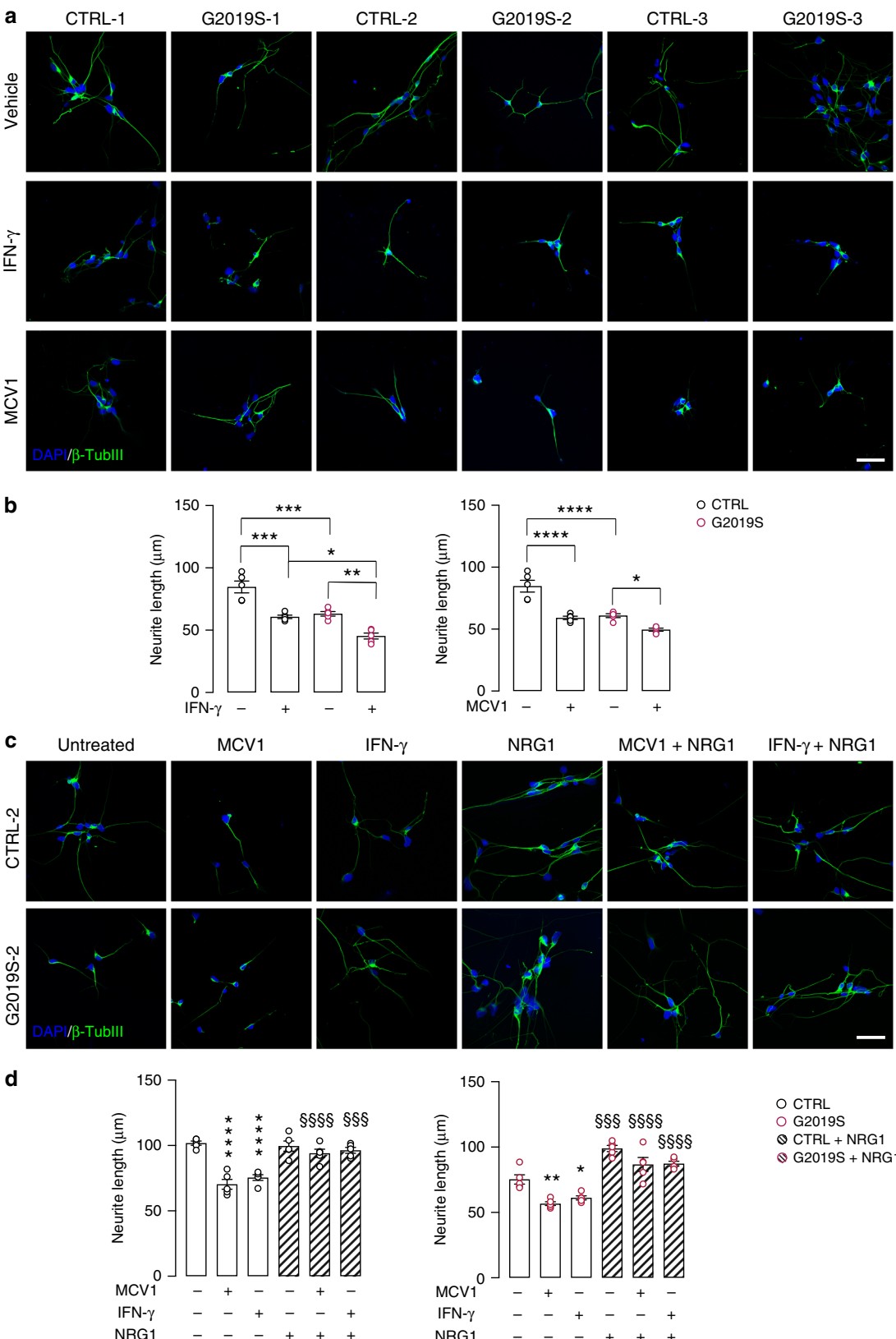

in G2019S microglia upon LRRK2 kinase inhibition (Supplementary Fig. 9B).

Next, we further investigated the impact of LRRK2 on the metabolic profile of LPS-stimulated microglia. To this end, we examined the extracellular acidification rate (ECAR) using a Seahorse Analyzer, which revealed a defect in the LPS-induced glycolytic switch in *LRRK2* KO microglia and controls upon LRRK2 kinase inhibition (Fig. 6g). Interestingly, G2019S microglia were more metabolically active upon LPS stimulation (Fig. 6g). Such differences in metabolic states among the different cell lines were confirmed using IFN-γ, which also increases glycolysis[43] (Fig. 6g). To confirm the impact of *LRRK2* G2019S on the proinflammatory

**Fig. 5 IFN-γ reduces neurite outgrowth by decreasing NFAT signaling. a** Representative images of β-III tubulin immunostaining (green) showing neurite elongation in *LRRK2* G2019S and isogenic control iPSC-derived neurons after treatment with vehicle, 500 nM MCV1 (NFAT inhibitor) for 24 h, or 200 IU/mL IFN-γ for 24 h. Blue, DAPI staining. Scale bar, 50 μm. **b** Quantification of neurite elongation is shown (mean ± SEM, two-way ANOVA, Bonferroni post hoc, left panel: ***$P$ = 0.0005, 0.0001 in sequence, **$P$ = 0.0029, *$P$ = 0.0103, right panel: ****$P$ < 0.0001, *$P$ = 0.0477; $n$ = 5 independent experiments). **c** Immunostaining for β-III tubulin (green) showing neurite elongation in *LRRK2* G2019S and isogenic control iPSC-derived neurons after treatment with vehicle, 500 nM MCV1 for 24 h, or 200 IU/mL IFN-γ for 24 h with or without prior treatment with 200 ng/mL NRG1 for 24 h. Scale bar, 50 μm. **d** Quantification of neurite elongation is shown (mean ± SEM, one-way ANOVA, Bonferroni post hoc; left panel: for vehicle vs MCV1/IFN-γ treatment ****$P$ < 0.0001; for NRG1 treatment, treated vs untreated §§§§$P$ < 0.0001, §§§$P$ = 0.0005; $n$ = 5 independent experiments), right panel: for vehicle vs MCV1/IFN-γ treatment **$P$ = 0.0032, *$P$ = 0.0436; for NRG1 treatment, treated vs untreated §§§§$P$ < 0.0001, §§§$P$ = 0.0002; $n$ = 5 independent experiments).

profile, we examined the response of microglia to other TLR agonists that elicit an IFN type II response. To this end, isogenic control, *LRRK2* G2019S, and *LRRK2* KO microglia were treated with the viral mimic polyinosinic:polycytidylic acid (poly[I:C],10 μg/mL), and production of IL-1β was assessed by ELISA. Levels of IL-1β were significantly decreased in poly(I:C)-stimulated *LRRK2* KO microglia compared to controls. Poly(I:C)-treated *LRRK2* G2019S microglia showed a non-significant increase in IL-1β production compared to isogenic controls (Fig. 6h).

Finally, iPSC-derived neurons were exposed to conditioned media from LPS-activated microglia (microglial-conditioned media [MCM]). The findings showed that only MCM from *LRRK2* G2019S activated microglia contributed to an inflammatory environment that affected neurite elongation (Fig. 6i and Supplementary Fig. 10).

## Discussion

Besides their role in immune defense, both type I and type II IFNs contribute to physiological brain functions, such as regulation of neurogenesis and synaptogenesis[6,44]. However, increasing evidence indicates that dysregulation of IFN signaling also plays a role in aging and neurodegenerative disease processes[2,45].

With respect to PD, genetic and functional studies have identified a link between IFN-γ and disease[4,5,8,11,12]. There is also considerable evidence that PD-related genes may synergize with inflammatory catalysts in determining disease risk[13,46,47]. Notably, PD genes may not only increase the vulnerability of substantia nigra DA neurons to inflammation-induced degeneration, but could also contribute to the dysregulation of immune responses to infections and the enhancement of age-related immune dysfunction[48]. This connection is particularly relevant to *LRRK2*, the most common genetic determinant of familial and sporadic PD[14]. LRRK2 function extends beyond the brain; it is involved in inflammatory bowel diseases, infections, and cancers[49]. Consistent with the role of LRRK2 in infections, the *LRRK2* promoter region contains binding sites for IFN response factors, and IFN-γ robustly induces LRRK2 expression in a variety of immune cells[15–19].

Using a human iPSC-based model, we found that IFN-γ also induces LRRK2 expression in both healthy control neurons and LRRK2 PD neurons. Thus, IFN-γ may serve as the central regulator of LRRK2-mediated responses during infections, immunity, and neuroinflammatory processes in a cell-type-specific manner. Healthy control neurons responded strongly to IFN-γ, as shown by the upregulation of genes involved in cellular stress responses (*EIF2AK2*), apoptotic pathways (*IRF1*, *SOCS1*), brain development and axonal regeneration (*AKT3*, *SOCS1*), and inflammasome assembly (*EIF2AK2*). However, IFN-γ alone did not elicit neuronal loss, which aligns with its primary role in promoting cellular integrity during infections.

In addition, we found that IFN-γ suppressed AKT phosphorylation in human DA neurons, with the strongest effect observed in G2019S neurons. Interestingly, IFN-γ treatment also suppresses AKT signaling in human macrophages[50], suggesting that

neuronal IFN-γ signaling shares similarity with immune cell signaling. Previous work has implicated LRRK2 in AKT phosphorylation, and shown that disease-associated mutations reduce its interaction with and phosphorylation of AKT[51–53]. Here, we show that IFN-γ synergizes with the *LRRK2* G2019S mutation to suppress AKT phosphorylation. Corroborating the involvement of AKT in the regulation of NFAT activity[27,28], we found that LRRK2 negatively regulates NFAT activity in human macrophages and iPSC-derived neurons and microglia. These data support previous findings by Liu et al. showing that LRRK2 negatively regulates NFAT in murine macrophages[31].

Furthermore, we identified a role for IFN-γ as negative regulator of NFAT in human neurons. This effect is mediated by the IFN-γ-dependent negative regulation of AKT phosphorylation. Conversely, IFN-γ-dependent activation of the proteasome[30] leads to degradation of NFAT. It is likely that proteasome activation is mediated by the IFN-γ-induced increase in SOCS1 levels, which is known to activate the ubiquitin-proteasome pathway[54]. Importantly, the G2019S mutation yields even more pronounced effects on NFAT activity due to its inhibition of AKT phosphorylation, which thereby impairs NFAT translocation[27,28]. In the original paper by Liu et al., LRRK2 was shown to negatively modulate NFAT via NRON complex-mediated cytosol retention[31]. Using *LRRK2* G2019S neurons, we identified alternative mechanisms underlying this negative modulation. Specifically, we show that *LRRK2* G2019S negatively modulates NFAT shuttling through two distinct mechanisms: by altering $Ca^{2+}$ dynamics and, likely, via a microtubule-dependent pathway.

The main activation pathway of NFAT involves increasing intracellular $Ca^{2+}$ via cell-surface receptors, resulting in the activation of phospholipase C (PLC)-γ and $Ca^{2+}$ release from the ER[55]. We show that *LRRK2* G2019S leads to defects in ER $Ca^{2+}$ storage in neurons, attenuating $Ca^{2+}$ release from the ER in response to TPH. Interestingly, Korecka et al. found a significant decrease in expression levels of the ER $Ca^{2+}$ sensor STIM1, which links ER $Ca^{2+}$ store depletion and SOCE activation in G2019S neurons[39]. The authors also show a reduced functionality of SOCE as a possible consequence of the decreased STIM1 expression[39]. Taken together, *LRRK2* G2019S might interfere with store-operated $Ca^{2+}$ entry and subsequent NFAT activation. In addition, we examined the impact of chronic IFN-γ treatment on intracellular $Ca^{2+}$ dynamics. While acute IFN-γ treatment is known to induce $Ca^{2+}$ transients[56], our data reveal that prolonged IFN-γ treatment depletes $Ca^{2+}$ in ER stores. This finding is in line with the reduced NFAT nuclear localization. The LRRK2-dependent role of $Ca^{2+}$ dynamics in NFAT activation was further supported by rescue experiments using NRG1, which positively regulates NFAT activation[40].

Moreover, we revealed an effect of LRRK2 on the microtubule network using super-resolution microscopy. We provide evidence that alterations caused by the *LRRK2* G2019S mutation in microtubules may affect NFAT nuclear translocation and activation. LRRK2 plays a physiological role in cytoskeletal organization, and pathogenetic variants alter microtubule stability with

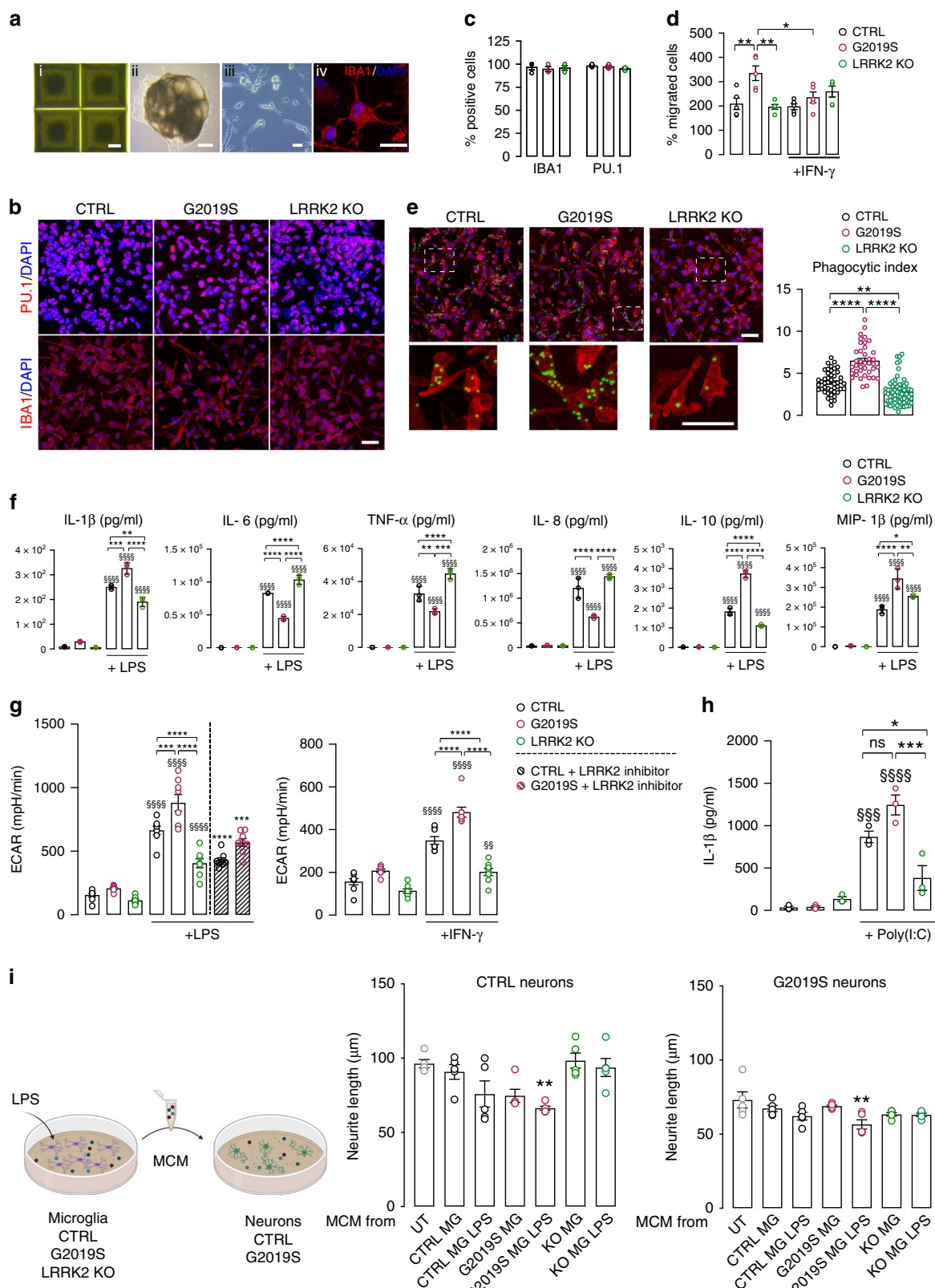

a cell-type-specific functional impact[35]. Combining dSTORM with SIFNE, a recent computational tool for the extraction of microtubule filaments from superresolution microscopy images[36], we obtained information about individual microtubule filaments and extracted the *LRRK2* G2019S microtubule network. We show that *LRRK2* G2019S leads to an increased complexity of the

microtubule network in the periphery of the cell in neurons, and that stabilizing microtubules partially rescues the LRRK2-mediated defects in NFAT shuttling.

Using iPSC-derived microglia, we found that *LRRK2* G2019S also leads to the defective nuclear translocation of another key transcription factor, NF-κB p65, which aligns with previous

**Fig. 6 LRRK2 regulates the function of human iPSC-derived microglia. a** Overview of microglia generation protocol using iPSC-derived embryoid bodies (EBs). Representative bright-field images of (i) day 3 and (ii) mature EBs and (iii) ramified iPSC-derived microglia. Scale bars, 50, 100, and 20 μm respectively. (iv) Confocal image of iPSC-derived microglia immunostained for IBA1 (red) and nuclear staining DAPI (blue). Scale bar, 20 μm. **b–c** Confocal images (**b**) of iPSC-derived microglia immunostained for PU.1 (red, upper), IBA1 (red, lower panels), and nuclear staining DAPI (blue). Scale bar, 20 μm. In (**c**) corresponding quantification (mean ± SEM, $n = 3$ independent experiments). **d** Quantification of migrated iPSC-derived microglia upon ATP stimulation with or without prior IFN-γ treatment, normalized over unstimulated control microglia (mean ± SEM, two-way ANOVA, Bonferroni post hoc; **$P = 0.0050$, 0.0017 in sequence, *$P = 0.0436$; $n = 5$ independent experiments). **e** Representative confocal images (left) and quantification (right) of microglia phagocytic capacity (green: latex fluorescent beads, red: IBA1). Scale bars, 20 μm; (mean ± SEM, one-way ANOVA, Bonferroni post hoc, ****$P < 0.0001$, **$P = 0.0011$; $n = 45$ control, $n = 39$ G2019S, $n = 67$ LRRK2 KO cells examined over three independent experiments). **f** Concentration of differentially secreted cytokines upon LPS treatment (mean ± SEM, two-way ANOVA, Bonferroni post hoc, treated vs untreated, §§§§$P < 0.0001$; compared to LPS-treated: ****$P < 0.0001$, ***$P = 0.0001$, 0.0006 in sequence, **$P = 0.0015$, 0.0017, 0.0056 in sequence, *$P = 0.0432$; $n = 3$ independent experiments). **g** Metabolic analysis of extracellular acidification rate (ECAR) in LPS-treated microglia with or without prior LRRK2 kinase inhibition (left) or IFN-γ-treated microglia (right) (mean ± SEM, two-way ANOVA, Bonferroni *post hoc*, treated vs untreated, §§§§$P < 0.0001$, §§$P = 0.0044$ compared to LPS- or IFN-γ-treated: ****$P < 0.0001$, ***$P = 0.0007$; compared to Mli-2-treated: two-tailed t-test, ****$P < 0.0001$, ***$P = 0.0006$; $n = 8$). **h** Concentration of secreted IL1-β upon poly(I:C) treatment (mean ± SEM, two-way ANOVA, Bonferroni post hoc, treated vs untreated, §§§§$P < 0.0001$, §§§$P = 0.0002$; compared to treated: ***$P = 0.0001$, *$P = 0.0201$; ns $P = 0.1097$; $n = 3$). **i** Schematic diagram showing treatment of neurons with either untreated or LPS-treated microglial-conditioned medium (MCM). On the right, quantification of neurite elongation in control and G2019S neurons (mean ± SEM, one-way ANOVA, Bonferroni post hoc, **$P = 0.0078$, 0.0051 in sequence, compared to UT; $n = 5$ independent experiments).

findings in LRRK2 G2019S neurons[57]. Interestingly, NF-κB nuclear shuttling is also regulated by microtubule-dependent transport[58]. Therefore, perturbations of the microtubule network observed in LRRK2 models may lead to a general disruption of the microtubule-dependent transportation and nuclear shuttling of key transcription factors. While it is well established that calcium interacts with and regulates the cytoskeleton, including microtubule dynamics, several works now link microtubules to SOCE, whereby microtubules have been shown to promote SOCE signaling[59]. Given the crucial role of SOCE in NFAT shuttling, LRRK2-related changes in microtubules may affect calcium handing and NFAT activity. Finally, we identified cell-type-specific effects of reduced NFAT translocation. Consistent with the role of calcineurin/NFAT signaling in the regulation of axon outgrowth[37], we show that NFAT signaling is involved in neurite outgrowth in human neurons, and that IFN-γ interacts with LRRK2 G2019S to inhibit NFAT activity and reduce neurite length. Therefore, our results provide further mechanistic explanations of the neurite growth defects that were observed in LRRK2 G2019S neurons[25]. Nonetheless, other NFAT-independent pathways may be involved in IFN-γ-induced neurite shortening. Furthermore, it is important to note that both NFAT and LRRK2 are involved in synaptic plasticity, also beyond the DA system[60,61].

To further examine the role of NFAT signaling in immune cells, we employed human macrophages and human iPSC-derived microglia. Whether adult microglia express LRRK2 remains controversial; our data show that LRRK2 is expressed by human microglia at basal conditions and that its expression is upregulated by IFN-γ. In addition, we show that human microglia produce IFN-γ when stimulated by LPS. However, we cannot exclude the possibility that the in vitro culture conditions of our study contribute to the upregulated LRRK2 expression and cytokine production. In line with its known role in cytoskeletal remodeling, IFN-γ stimulation reduced the motility of microglia; however, this effect was significant only in LRRK2 G2019S microglia, suggesting that PD-associated LRRK2 mutations modulate immune responses to IFN-γ. Using iPSC-derived microglia, we show that G2019S microglia display increased phagocytic activity, which is an NFAT-independent function[62]. In line with our findings in iPSC-derived neurons, we found that blocking LRRK2 activity led to increased NFAT nuclear localization in innate immune cells (human THP-1 macrophages and iPSC-derived microglia). Conversely, LRRK2 G2019S was associated with decreased nuclear NFAT. Interestingly, our data show

that decreased NFAT nuclear localization is already present in G2019S neurons at basal levels, whereas nuclear localization in immune cells is induced upon stimulation, indicating a cell-type-specific behavior that likely depends on cell-type-specific calcium dynamics. For example, specialized L-type $Ca^{2+}$ channels are known to regulate NFAT3 translocation and function in hippocampal neurons[63]. It is also interesting to note that $Ca^{2+}$ signaling differentially regulates key effector functions in adaptive and innate immune cells (macrophages and dendritic cells)[64].

Upon stimulation with LPS, which is known to promote NFAT nuclear translocation in microglia[42], we detected LRRK2- and NFAT-dependent changes in cytokine production. Specifically, IL-6, TNF-α, IL-8, and MCP-1 production were downregulated in G2019S microglia but upregulated in LRRK2 KO microglia. However, G2019S microglia showed increased production of IL-10, IL-1β, IL-12p70, and MIP-1β upon stimulation, suggesting that mechanisms other than NFAT regulate activation and cytokine responses in LRRK2 KO and G2019S microglia. In this respect, we show that LRRK2 G2019S impairs the nuclear translocation of the NF-κB p65 subunit upon LPS and IFN-γ stimulation. Other than its well-known role in proinflammatory gene expression, NF-κB has important anti-inflammatory and immune-suppressive functions. Furthermore, the inhibition of NF-κB causes cellular metabolic reprogramming towards glycolysis[65]. Interestingly, we found that LRRK2 regulated microglia activation by interfering with the metabolic switch toward glycolysis that normally occurs in LPS- and IFN-γ-activated macrophages.

Taken together, these data suggest that LRRK2 has an important function in regulating microglial activation through multiple, NFAT-independent mechanisms. Indeed, key innate immune functions, such as phagocytosis and immune cell metabolism, are NFAT-independent[62]. An intriguing hypothesis would be that DA neurons are selectively vulnerable to age-related and inflammatory challenges based on their $Ca^{2+}$ handling properties. Importantly, LRRK2 G2019S microglia activation provokes neurite shortening in conditioned media experiments, pointing towards a neurotoxic effect of LRRK2-driven immunological changes in PD. These findings have important consequences, as they highlight the possibility of modulating microglial function by external metabolic interventions. Given that the glycolytic switch underlies reactive phenotypes, using small molecules and metabolic intermediates to modulate glycolysis may repolarize microglia towards a less inflamed phenotype.

Increased IFN-γ levels have been documented in PD patients[8–10], although the source of IFN-γ in the aged and PD brain remains to

be identified. Our work shows that LRRK2 does not influence IFN-γ production in microglia, suggesting that microglial cells are not the primary source of IFN-γ. Aging is accompanied by immune dysfunction and a paradoxical increase in cytokines in several tissues, including the brain[48]. A recent study on mice demonstrated a clonal T-cell expansion within the brain of old mice with high levels of IFN-γ, suggesting a role for T-cell-derived IFN-γ in the age-dependent decline of brain function[66]. Given the presence of T-lymphocytes in PD brains and the growing evidence pointing towards a role for T-cell responses in disease pathogenesis[67,68], T-cells could be the main source of IFN-γ, which, in turn, induces LRRK2 expression in neurons and microglia, leading to neuronal damage and inflammatory reactions. Interestingly, a recent paper reported high α-syn-specific T-cell responses in PD patients at preclinical stage[69]. This early T-cell activation may be associated with aging and additional inflammatory events encountered over a lifetime (i.e., infections). Since reactive T-cells produce significant amounts of IFN-γ, these cells might serve as the primal source of IFN-γ production, driving immune responses, including the metabolic reprogramming and activation of microglia, as well as neuronal dysfunction, which is partially mediated by NFAT-related mechanisms. Thus, further studies are needed to dissect the signature of distinct subpopulations of T-cells responsible for early immune activation in PD. As indicated by our data and data from the literature, LRRK2 also modulates other essential microglial functions, including motility and phagocytosis. Interestingly, MHC-I is highly expressed by substantia nigra DA neurons and locus coeruleus noradrenergic neurons[70]. Furthermore, primary DA murine neurons and human embryonic stem cells (hESCs) are more susceptible to MHC-I induction by IFN-γ than other neuronal populations[70]. These mechanisms would render catecholaminergic neurons selective targets for immune-mediated cell death, as observed in viral parkinsonism[6]. In this scenario, LRRK2 G2019S individuals may be even more vulnerable to IFN-γ-driven mechanisms.

In summary, our work shows that the G2019S mutation sensitizes neurons to IFN-γ signaling, serving as a potential direct link between inflammation and neurodegeneration in PD (Fig. 7). We have uncovered a cell-type-specific role for the synergistic effect of IFN-γ and LRRK2, which may increase PD risk through different age-related signatures that converge on the immune system. Importantly, given the role of LRRK2 in sporadic PD[71,72], the relevance of our work extends beyond patients with the LRRK2 G2019S mutation, and suggests that the IFN-γ/LRRK2 axis may be a potential target for intervention in acute or chronic states of neuroinflammation in genetic and sporadic PD.

## Methods

**Differentiation of NPCs into midbrain DA neurons.** All cells used in the study were derived from patients who signed an informed consent form[25]. The ethics committee of the Medical Faculty at the University Hospital Tübingen (Ethikkommission der Medizinischen Fakultät am Universitätsklinikum Tübingen) approved the protocol prior to performing the experiments. Patient-derived LRRK2 G2019S and isogenic gene-corrected control iPSC-derived NPCs were previously generated, and NPC differentiation into DA neurons was accomplished according to a previously published protocol[24]. In brief, NPCs were passaged with Accutase (Sigma-Aldrich) every 5–6 days on Matrigel-coated plates at a 1:5–1:10 ratio in an expansion medium containing N2/B27, 150 μM ascorbic acid (AA; Sigma-Aldrich), 3 μM CHIR 99021 (CHIR; Axon Medchem), and 0.5 μM purmorphamine (Merck Millipore). NPCs were plated for further differentiation and upon reaching 70% confluency, the expansion medium was replaced by N2/B27 medium supplemented with 200 μM AA, 100 ng/mL FGF8 (PeproTech), and 1 μM purmorphamine; the medium was changed every other day. On day 8, maturation medium containing N2/B27 medium with 20 ng/mL BDNF (PeproTech), 500 μM dbcAMP (Applichem), 1 ng/mL TGFβ3 (PeproTech), 20 ng/mL GDNF (PeproTech), 200 μM AA, and 0.5 μM purmorphamine was applied to the cells for a duration of two days. After switching to the maturation medium, confluent cultures were detached with Accutase and replated at a 1:3 ratio on Matrigel-coated plates. Cells were kept for two weeks in maturation medium before performing experiments. Cell lines were routinely assessed by Sanger sequencing for cell line identity and for mycoplasma

(every two months). Immunostaining for β-III tubulin and TH was also performed for a qualitative evaluation of neuronal differentiation. Only cultures that effectively differentiated into DA neurons were used for experiments. Differentiation into cortical neurons was performed using N2/B27 medium supplemented with BDNF, GDNF, dbcAMP, and AA[73].

**Reagents.** Where indicated, cells were treated with recombinant human IFN-γ (R&D Systems), LPS (LPS from E. coli O111:B4), poly(I:C) (both Invivogen), ionomycin, MG132, colchicine, PMA (all from Sigma-Aldrich), LRRK2-IN-1 and GSK2578215A (both Tocris), Mli-2 (Abcam), MCV1 (HPVIVIT, Calbiochem), recombinant human NRG1 (BioLegend), and PTX (STEMCELL Technologies).

**Generation of LRRK2 KO iPSCs.** LRRK2 KO iPSCs (clone #1) were generated using zinc-finger nucleases (ZFNs; Sigma-Aldrich). Cells were transfected with 2 μg of each ZFN construct, as well as 2 μg linearized targeting vector harboring a premature stop codon in LRRK2 exon 41 and a neomycin resistance cassette, using Amaxa Nucleofector II (Lonza), Nucleofection Solution for human stem cells II (Lonza), and program B-16, according to the manufacturer's instructions. Cells were replated onto mitomycin-treated CF-1 mouse embryonic fibroblasts (MEF) (Thermo Fisher) feeder-coated plates in hESC medium supplemented with 10 μM ROCK inhibitor (Ascent Scientific). After colony formation, selection for homologous recombination was performed by adding 50 μg/mL G418 (PAA). Resistant colonies were selected and clonally expanded. A second step of nucleofection was performed with 2 μg of the same homologous construct harboring a blasticidin resistance cassette. The selection was performed with 100 μg/mL of blasticidin (InvivoGen), and resistant colonies were expanded on MEF feeder-coated plates. The presence of the stop codon in exon 41 was validated by Sanger sequencing (primers FW: GCACAGAATTTTTGATGCTTG; RV: GAGGTCAGTGGTTATC-CATCC). LRRK2 KO clones (clone #2 and #3) were generated from a newly generated viral-free control iPSC line (CTRL-4). For iPSC generation, human fibroblasts were cultured in fibroblast culture medium: DMEM high glucose (Life Technologies) plus 10% FBS (Life Technologies). Reprogramming was performed by nucleofection with the episomal plasmids pCXLE-hUL, pCXLE-hSK, and pCXLE-hOCT4, as described by Okita et al.[74]. Pluripotency was assessed by immunostaining for pluripotency genes (OCT4, SOX2, TRA-1-60, TRA-1-80) and EB-based differentiation. To verify the absence of integration of the reprogramming plasmids, RT-PCR was performed with plasmid-specific primers[74]. The genomic integrity of selected clones was assessed by G-banded karyotyping and high-density genotyping using the Illumina HumanOmni2.5-8 array. Copy number analysis was performed using the cnvPartition plugin (Illumina). Synthetic sgRNAs targeting LRRK2 exon 3 were purchased from Synthego (sgRNA1: CAA UCA UUU CCA UCA UCC UG; sgRNA2: AAA UUA AUA GAA GUC UGU CC). Nucleofection of 180 pmol sgRNA1, 180 pmol sgRNA2, and 20 pmol Cas9 nuclease (sgRNA to Cas9 nuclease ratio used was 9:1) was performed with Amaxa 2D Nucleofector (program B-016; Lonza Bioscience). After recovery, iPSCs were clonally expanded and the genomic deletion assessed by PCR (FW: GGTGGGTTG GTCACTTCTGT; RV: ACAGGAGTAGCCTTGGATTGC) and Sanger sequencing (FW: ACGGCTTGCTTTTGTTTCTGG; RV: GCAAACACAGTGTATCAAGG GA). The screening of possible off-target effects was performed using crispr.mit.edu and crispr.cos.uni-heidelberg.de; no exonic off-target effect was predicted. The genomic integrity of selected clones was assessed by G-banded karyotyping and high-density genotyping using the Illumina HumanOmni2.5-8 (Omni2.5) array. iPSCs were kept in culture in hESC medium consisting of Gibco KnockOut DMEM (Life Technologies), 20% Gibco KnockOut Serum Replacement (Life Technologies), 1% penicillin/streptomycin (P/S, Merck Millipore), 1% Gibco Non-Essential Amino Acids (Life Technologies), 500 μM β-mercaptoethanol (Sigma-Aldrich), and 1% Gibco GlutaMAX Supplement (Life Technologies) supplemented with 10 ng/ml FGF2 (PeproTech). iPSCs were maintained by splitting onto MEF feeder-coated plates with medium supplemented with 10 μM ROCK inhibitor Y-27632 2HCl (Selleck Chemicals).

**Differentiation of human iPSCs into microglia.** iPSCs were differentiated into microglia following an established protocol[75] with minor changes. In brief, embryoid bodies (EBs) were formed using AggreWell™800 (STEMCELL Technologies), and cultured in mTeSR1 (STEMCELL Technologies) with bone morphogenetic protein 4 (BMP4, 50 ng/mL; ImmunoTools), vascular endothelial growth factor (VEGF, 50 ng/mL; ImmunoTools), and stem cell factor (SCF, 20 ng/mL; ImmunoTools) for four days with 75% medium change daily. On day 4, EBs were collected and transferred into 6-well cell culture plates (12–16 EBs/well) in X-VIVO 15 (Lonza) supplemented with IL-3, (25 ng/mL; ImmunoTools), macrophage colony-stimulating factor (M-CSF, 100 ng/mL; ImmunoTools), 2 mM GlutaMAX (Thermo Fisher), 1% P/S (Merck Millipore), and 0.055 mM β-mercaptoethanol (Sigma-Aldrich), with medium change weekly. After 3–4 weeks, floating cells were collected and seeded at a concentration of 100,000 cells/cm² on Matrigel-coated plates in Advanced DMEM/F-12 (Life Technologies) supplemented with N2 (Thermo Fisher), GlutaMAX, P/S, β-mercaptoethanol, M-CSF (100 ng/mL), IL-34 (100 ng/mL; PeproTech), and GM-CSF (10 ng/mL, ImmunoTools) with medium change twice a week.

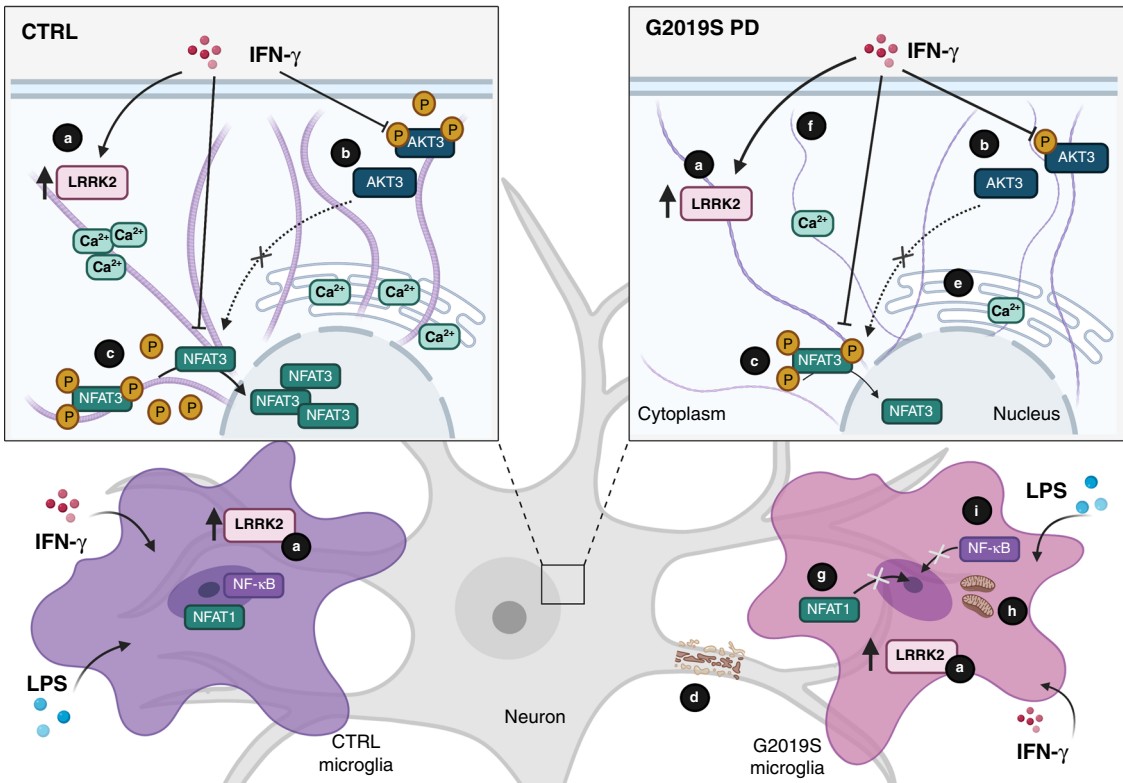

**Fig. 7 Interferon-γ signaling synergizes with LRRK2 in human neurons and microglia.** IFN-γ response is conserved in DA neurons, and IFN-γ induces LRRK2 expression in DA neurons and microglia (**a**). The PD-associated mutation *LRRK2* G2019S sensitizes DA neurons to IFN-γ by decreasing AKT phosphorylation (**b**) and suppressing NFAT nuclear shuttling (**c**) that, in turn, leads to defects in neurite outgrowth (**d**). LRRK2-dependent defects of NFAT translocation are linked to defects in calcium buffering capacity (**e**) and possibly to an increased complexity of the microtubule network (**f**). Defects in NFAT shuttling are also observed in *LRRK2* G2019S microglia (**g**), which display a more activated phenotype upon stimulation. *LRRK2* G2019S modulates microglia activation by interfering with the metabolic switch toward glycolysis that normally occurs upon mononuclear phagocyte activation (**h**) and impairing NF-κB p65 nuclear translocation upon stimulation with IFN-γ or LPS (**i**).

**Experiments with microglial-conditioned medium (MCM)**. The conditioned medium was collected after 24 h from untreated or 100 ng/mL LPS-activated microglia. MCM was then filtered, diluted 1:1 with neuronal medium, and applied to neurons. Cells were fixed after 48 h and subjected to neurite length assay.

**THP-1 experiments**. Human THP-1 cell lines were purchased from Sigma-Aldrich and cultured in RPMI 1640 (Life Technologies), 2 mM GlutaMAX, and 10% FBS. THP-1 cells were differentiated into macrophages with 25 ng/mL PMA (Sigma) for 48 h. For *LRRK2* KD experiments, the following high-titer lentiviruses were purchased from Sigma MISSION Library: TRCN0000021462 $10^8$ TU vector pLKO.1, TRCN0000021460 $10^8$ TU vector pLKO.1, and SHC002V non-target control. Lentiviral infection of THP-1 macrophages was performed with spinfection at an MOI of 10.

**NFAT Reporter—HEK293 cell line and Luciferase assay**. The NFAT Reporter— Hek293 cell line was purchased from BPS Bioscience and cultured in DMEM (Merck) supplemented with 10% (vol/vol) FBS (Gibco) and 400 μg/mL of G418. Where specified, cells were transfected with plasmid DNA (SF-tagged wt *LRRK2* or SF-tagged *LRRK2* G2019S) using 1 mg/mL polyethylenimine (PEI) solution (2.5 μg of plasmid DNA/μL; Polysciences). Where specified, cells were treated with 10 μM colchicine for 30 min at 37 °C or 100 nM PTX for 24 h. Cells were stimulated with 40 ng/mL PMA (Sigma-Aldrich) and 1 μM ionomycin (Sigma-Aldrich) for 24 h. Luciferase activity was determined using the Luciferase Assay System (Promega) according to the manufacturer's protocol and measured with a TriStar$^2$ S LB 942 Multimode Microplate Reader (Berthold).

**Western blotting**. Proteins were kept on ice and extracted using NP40 lysis buffer (Tris-buffered saline [TBS] plus 0.5% Nonidet P-40, pH 7.4) containing protease and phosphatase inhibitors (Roche) following centrifugation for 15 min at 4 °C. The protein concentration of the supernatant was determined by BCA (Pierce, WI, USA). In total, 30–100 μg of the protein lysate was loaded onto a polyacrylamide gel (7.5–15% gel depending on the molecular weight of the protein) and transferred onto a PVDF membrane (Millipore, MA, USA). Blots were blocked with 5% milk powder or 5% BSA in TBS containing 0.1% Tween-20 and incubated with primary

antibodies in blocking solution (Western Blotting Reagent, Roche) for 1 h. Blots were then incubated overnight at 4 °C with the primary antibody of interest. The appropriate species of HRP-conjugated secondary antibody (all Sigma-Aldrich) resuspended in 5% milk or 5% BSA was then applied to the membrane for 1 h at room temperature.

Nuclear-cytosolic separation was performed using NE-PER Nuclear and Cytosolic Extraction Reagent Kit (Thermo Fisher) according to the manufacturer's instructions. 10 μg of protein was loaded on Tris-HCL NuPAGE gradient 4-12% gradient gel, and the membrane was probed for cytoplasmic and nuclear compartment markers. Proteins were visualized with Amersham ECL Western Blotting Detection Reagent and Amersham Hyperfilm (both GE Healthcare, IL, USA). ImageJ software (v. 1.52a) was used for densitometric analysis.

The primary antibodies used were: rat monoclonal anti-LRRK2 24D8 (2 μg); rat monoclonal anti-LRRK2 1E11 (4 μg; both were kindly gifted by C. Johannes Gloeckner); rabbit anti-LRRK2 (UDD3 30[12]; 1:500; Abcam, #ab133518); rabbit anti-LRRK2 (MJFF2 [c41-2]; 1:1000; Abcam, # ab133474); rabbit anti-AKT3 (1:1000; Cell Signaling Technology, #4059); rabbit anti-NFAT3 (23E6; 1:1000; Cell Signaling Technology, #2183); rabbit anti-PARP (1:8000; Cell Signaling Technology, #9542); rabbit anti-HSP90 (1:15000; Enzo, #ADI-SPA-836F); mouse anti-β-Actin (1:20000; Sigma, #A5441); rabbit anti-AKT (1:1000; Cell Signaling Technology, #9272); rabbit anti-phospho-AKT (Ser473; 1:1000; Cell Signaling Technology, #4060).

**Immunocytochemistry and image analysis**. Cells were fixed in 4% paraformaldehyde (PFA) in PBS (w/v) for 15 min, washed with PBS, and blocked with PBST (PBS with 0.1% Triton X-100) with 10% normal goat serum (NGS) for 30 min. The cells were then incubated overnight at 4 °C with the primary antibody of interest diluted in blocking solution. The appropriate species of Alexa Fluor 488/ 568/647-conjugated secondary antibody (Invitrogen) resuspended in PBST with 1% NGS was then applied to the cells for 1 h at room temperature.

The primary antibodies were: rabbit anti-NFAT1 (D43B1; 1:1000; Cell Signaling Technology, #5861); rabbit anti-phospho-NFκB p65 (Ser536; 1:500; Thermo Fisher, #MA5-15160); rabbit anti-NFAT3 (phospho-S676; 1:250; Biorbyt, #orb256717); mouse anti-β-III tubulin (TUBB3; 1:1000; Covance, #MMS-435P); rabbit anti-tyrosine hydroxylase (1:500; Pel-Freez, #P40101-150); mouse anti-Spi1/PU.1

(1:100; BioLegend, #658002); rabbit anti-Iba1 (1:2000; Wako Chemicals, 016-20001).

For NRG1 experiments, neuronal cultures were treated with 200 ng/mL recombinant human NRG1 for 48 h. For NFAT inhibition experiments, cells were treated with 500 nM MCV1 for 24 h. Images were acquired with a Zeiss LSM 510 confocal microscope and analyzed with ImageJ. For quantification of neuronal markers, images were acquired at a ×40 magnification, and at least 1000 cells were counted for each cell line. The mean fluorescent intensity of phospho-NFAT3 and the nuclear and cytoplasmic fluorescence were quantified by ROI analysis. NFAT1 levels in the cytosol and nucleus regions of interest (ROI) of identical size were drawn in the cytosol and nucleus. Nuclear localization was confirmed by co-staining with the nuclear dye DAPI and PU.1 immunostaining was employed to confirm microglia identity.

The evaluation of NFAT and phospho-NFκB-p65 was performed using the Intensity Ratio Nuclei Cytoplasm tool for ImageJ and plotted as the percentage of nuclear localization. Images were blindly acquired at x63 magnification, and 200–300 cells were scored for each condition. For neurite length analysis, neurons were plated at a low density (25,000 cells/cm$^2$) and fixed for immunostaining after 48 or 72 h, depending on the treatment. Images were analyzed using the Fiji plugin Simple Neurite Tracer.

**Cytokine analysis**. Levels of IFN-γ, IL-1β, IL-1Ra, IL-4, IL-6, IL-8, IL-10, IL-12p70, MCP-1, MIP-1β, TNF-α, and VEGF were determined using a set of "in-house developed" Luminex-based sandwich immunoassays, each consisting of commercially available capture and detection antibodies and calibrator proteins. Samples were diluted to at least 1:4 or higher to receive results below the upper limit of quantification. After incubation of the pre-diluted samples or calibrator protein with the capture-coated microspheres, beads were washed and incubated with biotinylated detection antibodies. Streptavidin-phycoerythrin was added after an additional washing step for visualization. For control purposes, calibrators and quality control samples were included on each microtiter plate. All measurements were performed on a FLEXMAP 3D® analyzer system using xPONENT® 4.2 software (Luminex, Austin, TX, USA). MasterPlex QT v. 5.0 was used for data analysis. IL-1β levels after poly(I:C) stimulation were measured using Human IL-1β/IL-1F2 DuoSet ELISA (R&D Systems).

**Intracellular bead-based multiplex assay**. Phospho-AKT3 (Ser473) and total AKT3 levels were measured using the MILLIPLEX MAP Total/Phospho-Akt3 2-plex Panel (EMD Millipore #48-633MAG) according to the manufacturer's instructions.

**Quantitative RT-PCR**. mRNA isolation and reverse transcription reaction were performed with RNeasy Mini Kit and with QuantiTect Reverse Transcription Kit (both from Qiagen), respectively, according to manufacturer's instructions. QuantiTect SYBR Green Kit (Qiagen) and a ViiA 7 RT-PCR System (Applied Biosystems) were used for quantitative PCR. The expression level of each gene was normalized to the levels of the housekeeping genes ribosomal protein large P0 (*Rplp0*), hydroxymethylbilane synthase (*HMBS*), or β2-microglobulin (*B2M*). The $2^{-\Delta\Delta CT}$ method was used to calculate fold changes in gene expression, based on housekeeping genes and biological reference samples for normalization. Primer sequences are provided in Supplementary Table 1.

**Seahorse XFe96 metabolic flux analysis**. ECAR was analyzed using an XF$^e$96 Extracellular Flux Analyzer (Seahorse Biosciences). iPSC-derived microglia were grown on V3 PS XF microplates (Seahorse Biosciences) at a density of 70,000 microglia per well for 14 days. Measurement of ECAR was performed using a Seahorse XF$^e$96 Analyzer (Agilent) in freshly prepared medium, consisting of phenol-free DMEM supplemented with 1 mM glutamine and pH adjusted to 7.4. Glycolytic function was evaluated after subsequent injection of 10 mM glucose, 10 μM oligomycin, and 50 mM 2-deoxy-D-glucose (all Sigma-Aldrich); for each condition, three measurements, each with 5-min duration, were performed. The values were normalized to cell number by counting DAPI-stained nuclei using a high-content cell analyzer (BD Bioscience, Pathway 855).

**Microglial cell motility assay**. Microglia were plated in the upper chamber of trans-well plates (5-μm-pore polycarbonate filters in 24-wells; Corning), containing adenosine triphosphate (ATP, 100 μM; Sigma-Aldrich) in the bottom chamber to induce chemotaxis. Wells containing medium without ATP were used as controls. After 4 h, cells were fixed with 4% PFA for 15 min at room temperature. Cells on the upper side of the filter were removed with a cotton swab. Cells on the bottom side of the chamber were counted under a light microscope, and the number of migrated cells was expressed as a percentage of the non-ATP-stimulated controls.

**Phagocytosis assay**. To assess the phagocytic capacity of microglia, cells were incubated with 0.01% (v/v) fluorescent latex beads (#L1030, Sigma-Aldrich) with a diameter of 1 μm for 90 min at 37 °C. Cells were washed three times with cold PBS and fixed with 4% PFA. After immunostaining, images were analyzed with ImageJ. The phagocytic index was calculated with the formula: (total number of engulfed beads) / (total number of IBA1-positive cells containing beads).

**Calcium live-cell imaging**. iPSC-derived neurons were loaded with Fura-2-AM (Invitrogen) for 40 min at 37 °C and 5% CO$_2$. The ratiometric recordings were performed in the recording medium by an upright fluorescence microscope (BX50WI, Olympus) equipped with a 20× water immersion objective (LUM-PLANFL, 20×/0.80 W, ∞ /0; Olympus), a polychromator (VisiChrome; TILL Photonics) and a CCD camera (Retiga R1, 1360 × 1024 pixels, 16-bit). During the recording, stacks (single-plane two-channel) of the cellular Fura-2 fluorescence at the focal plane were acquired at 2 Hz (λexc = 340 and 380 nm; Olympus U-MNU filter set, 40 ms exposure time, 8-pixel binning) using the VisiView software (TILL Photonics). The temperature of the recording chamber was set to 37 °C.

Baseline Ca$^{2+}$-activity was recorded for 20 min, consisting of 5 min baseline-recording and following 15 min of post-recording under different conditions by application of 500 nM of TPH (Sigma-Aldrich). Neuronal cultures were previously treated with 200 IU/mL IFN-γ or 200 ng/mL NRG1 for 24 h, where indicated. Extracellular recording medium contained 140 mM NaCl, 4 mM KCl, 2 mM CaCl$_2$, 1 mM MgCl$_2$, 4 mM glucose, and 10 mM HEPES.

For data analysis, the calcium-imaging ratio-stacks were generated by dividing the fluorescence images recorded at the excitation wavelengths of F340 and F380 (ImageJ, RatioPlus, https://imagej.nih.gov/). To detect cellular Ca$^{2+}$-values, selected cells were manually encircled by ROIs and the obtained ROIs coordinates were used to extract corresponding Ca$^{2+}$ traces from the ratio-stacks. The extracted traces were imported in Matlab (Matlab v.R2019b, Mathworks, USA) for further analysis. Next, the Ca$^{2+}$ traces were normalized and denoised using the Matlab in-built function sgolayfilt (order, framelen) with order=2 and framelen= 5. The baseline-value was generated as the mean of five values taken at each minute during the 5-min control-recording time. The agent-dependent increase in cellular Ca$^{2+}$ levels was calculated as the difference of peak-Ca$^{2+}$ value post-application and the predefined baseline value. The peak value and time were attained by the function [value, indices]=max(Ca2-trace) within the corresponding agent-application time.

**dSTORM super-resolution imaging**. Control and G2019S iPSC-derived neurons were treated with a freshly prepared microtubule stabilization buffer (80 mM PIPES, 1 mM MgCl2, 5 mM EDTA, pH 6.8 plus 0.5% Triton-X) for 10 s and then gently fixed with MetOH at –20 °C for 10 min. Fixed cells were washed with 1× PBS and incubated with blocking solution (5% NGS in PBS) for 30 min. Micro-tubules were stained using mouse anti-β-III tubulin monoclonal antibody (Abcam, Cambridge, UK) and Alexa Fluor 647-conjugated secondary antibody (Thermo Fisher) and imaged in a Nikon N-STORM 4.0 microscope.

Bright-field reference images were taken prior to dSTORM imaging and were used for ROI selection during the analysis. β-III tubulin labeling was confirmed using a Cy5 filter cube. Setup was controlled by NIS-Elements AR software (Nikon Instruments). Fluorescent light was filtered through a Cy5 cube (AHF; EX 628/40; DM660; BA 692/40) and Nikon Normal STORM cube (T660lpxr, ET705/72 m). Filtered emitted light was imaged with ORCA-Flash 4.0 sCMOS camera (Hamamatsu Photonics).

For epifluorescent widefield imaging, a fluorescent lamp (Lumencor Sola SE II) was used as a light source. dSTORM imaging was performed in total internal reflection fluorescence (TIRF) or highly inclined and laminated sheet microscopy (HiLo) mode with continuous 647-nm laser illumination (full power). For dSTORM, Nikon Normal STORM cube was used. Frame size was 128 × 128 pixels and image depth was 16-bit. For each dSTORM image, 20,000 frames were acquired at 33 Hz.

Image processing was performed with NIS-Elements AR software. Molecule identification settings were set to defaults for dSTORM analysis: minimum width = 200 nm, maximum width = 400 nm, initial fit width = 300 nm, max axial ratio = 1.3, and max displacement = 1. The minimum height for peak detection was set to 1000, and localization analysis was performed with overlapping peaks algorithm. dSTORM images were reconstructed with Gaussian rendering size of 10 nm in NIS Elements AR. Brightness and contrast were adjusted from 0 to 100 with Gaussian rendering intensity. Final dSTORM images were then exported as TIFF images. dSTORM images were further analyzed with the Matlab-based open-source software SMLM image filament network extractor (SIFNE), keeping the parameters constant across experiments[36]. Otsu's threshold was used to extract the binarized image, and filaments shorter than 0.2 μm were excluded from analysis.

**Statistical analysis**. Prism 8 (GraphPad Software, San Diego, CA, USA) was used to analyze the data. Statistical testing involved two-tailed Student's *t*-test, multiple *t*-test, one-way ANOVA, two-way ANOVA, followed by Bonferroni multiple comparison test or Kopalmogorov–Smirnov test, as indicated. Data are expressed as mean ± SEM. The significance level was set at $P < 0.05$.

**Reporting summary**. Further information on research design is available in the Nature Research Reporting Summary linked to this article.

## Data availability

Source data are provided with this paper.

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

## Acknowledgements

We thank Stefanie Kalb and Gabriele Di Napoli for the excellent technical support. We also thank Christian Johannes Gloeckner for sharing with us the LRRK2 antibodies. This work was supported by German Center for Neurodegenerative Disease (DZNE) and the Helmholtz Association Young Investigator Award (VH-NG-1123, to M.D.); the Marie Curie Career Integration Grant MC CIG304108 (M.D.), DFG research grant (DE 2157/2-1), COEN grant 4008 (M.D.); the Emmy Noether Programme of DFG (project number 317530061 to I.N.-S.), Hector Foundation (W.H.), and Tistou and Charlotte Kerstan Foundation (W.H.). Figure 7 was created with BioRender.com.

## Author contributions

M.D., S.D.C., D.I., and V.P. designed the experiments. S.D.C., D.I., and V.P. performed most of the experiments and analysed data; W.H., D.I., and V.P. designed, performed, and analysed data relative to Ca²⁺ measurements; C.Y. and D.C.S. generated and characterized *LRRK2* KO iPSCs; D.M., M.O., and M.J.P. performed experiments and analysed data; I.N. and A.A. designed and performed STORM experiments; V.P. and S.D.C. analysed STORM data; D.C.S. performed initial experiments on interferon signaling; R.P.C. performed microglia motility experiments; N.S. and M.J. performed Multiplex Elisa and Luminex experiments; T.G. provided patient samples; M.D. conceived and supervised the study, acquired funding, and wrote the manuscript with contribution from all authors.

## Funding

## Competing interests

The authors declare no competing interests.
