## [Peer Review File · Nature Communications]

Reviewers' comments:

Reviewer #1 (Remarks to the Author):

INF- γ -mediated mechanisms involved in neurodegeneration during Parkinson's disease (PD) is challenging and up-to-date subject, especially in light of growing body of evidence of involvement of inflammatory processes in PD pathology and of an association of PD risk with influenza infection and unbalanced Th1 cells (one of the main INF- γ producers). The link between LRRK2 and INF- γ is particularly intriguing, since it provides at least a part of the biological meaning of how the most frequent PD-associated mutation LRRK2 G2019S is mediating disease pathology.

While these findings are exciting news and add to our understanding of neuroinflammation in PD, the presentation of the data needs major improvements. In addition, the physiological relevance of these findings needs to be clearly explained. The data need to be rearranged to make sure, the reader can follow the flow of the manuscript. Moreover, a visual abstract of the finding would be helpful.

Major points:

1. Fig. 1 – How do the used INF- γ concentrations correlate with physiological concentration measured in serum or CSF samples of PD patients?

Is INF- γ -mediated LRRK2 increase specific to DA neurons? Why is the dose explicitly mentioned for the second paragraph of results but not for the first?

2. Fig. 2 – The level of overexpression of LRRK2 G2019S seems to be slightly higher in HEK cells than the overexpression of LRRK2 WT in Fig. 2A. Consequently, only slightly lower levels of NFAT luciferase after PMA/Ionomycin stimulation were determined in HEK cells overexpressing LRRK2 G2019S compared to LRRK2 WT HEK cells in Fig. 2B. Thus, it might be not an effect of the mutant LRRK2 form, but rather an effect of higher LRRK2 protein.

Moreover, if the PMA/Ion-mediated NFAT luciferase increase in LRRK2 G2019S HEK cells is not significantly different from the basic line (unstimulated HEK cells; Fig. 2B), how effective any other treatment including INF- γ treatment may be in reducing its activity? Maybe combining data on Fig. 2B and 2C would give more clear picture to the real reduction rate in different HEK cells by INF- γ .

3. Fig. 2D-G – it is hard to detect any NFAT nuclear localization in neurons on the provided pictures (Fig. 2D);

Would neurons need any additional stimulation for NFAT nuclear translocation similar to what was done for HEK cells or for neurons in the Suppl. Fig. 1H (Ionomycin treatment)?

The relevance of NFAT in neurons needs to be mentioned at this point (it is mentioned first much later in the pre-last part of the Results).

What was the reason for NFAT3 overexpression in iPSC-derived neurons? Instead, an evaluation of endogenous NFAT3 by means of, for example, immunofluorescence staining would firstly, stress a relevance of NFAT3 in neurons; and secondly, provide a physiological influence of INF- γ on NFAT3 neuronal localization.

What individual dots in E, F, G represent? If these are individual experiments (independent differentiation rounds?) as stated in the figure legend, that please explain how many cells were analyzed in each experiment.

4. LRRK2 KO iPSC lines (Suppl. Fig. 1E-F) are raising several questions:

- LRRK2 bands in left and right parts of the figure corresponding to CTRL-1 and CTRL-4, respectively, do have a different pattern and this needs to be explained;

- LRRK2 bands are different in CTRL-1+/+ samples on the left and right CTRL-1 WB pictures (and also different in the CTRL-4+/+ samples on the left and right CTRL-4 WB pictures): these differences must be explained;

- more detailed labeling of the Suppl. Fig. 1E could be helpful to better understand the figure message;

- LRRK2 protein signal in +/+ samples in the Suppl. Fig. 1E looks slightly different from the LRRK2 signal in Suppl. Fig. 1B: does this represent differences between iPSC-derived neurons (1B) and microglial cells (1E)? Please explain;

- LRRK2 KO iPSC-derived neurons or microglia cells must be characterized for at least neuronal and microglial markers in order to assess differentiation efficiency.

5. iPSC-derived microglia cells require more detailed characterization, which is completely omitted in the manuscript. First results from iPSC-derived microglial cells are presented "suddenly" in the Suppl. Fig. 1D-E without providing a clear rationale for using these cells to evaluate NFAT and LRRK2 expression and without any characterization of the generated microglia cells. Moreover, taking into account that total cell lysates were used to evaluate NFAT mRNA (Suppl. Fig. 1D) and LRRK2 protein (Suppl. Fig. 1E) expression, without knowing the efficiency of microglial cell differentiation and the frequency of microglial cells in the resulting cultures, it is hard to attribute respective signals to microglial cells.

6. Why in Suppl. Fig. 1F WB picture (on the left) corresponds to NFAT3 and the quantification (on the right) is performed on total NFAT? Please clarify or show total NFAT WB signals, which were quantified. Please clarify discrepancy between the legend to the Suppl. Fig. 1F-I stating "total NFAT3" and the text of the Results part (page 6) saying "total NFAT".

NFAT3 signal in Suppl. Fig. 1G-H looks different from NFAT3 signal in Suppl. Fig. 1F and 1I. Why NFAT3 signal looks different between total cell lysates (1F, 1I) and in cytosolic and nuclear fractions (1G-H).

Why in fractionation experiments presented in Suppl. Fig. 1H an additional Ionomycin stimulation was performed and in 1G not?

7. Fig. 4A-I – the results on microtubule network differences depending on LRRK2 mutation are very interesting, but shown only in HEK cells. The relevance of this phenomenon to human PD pathology could be strengthened by evaluation of microtubule state (stable vs. unstable) or microtubule networks in iPSC-derived neurons from LRRK2 G2019S PD patients compared to isogenic controls. Moreover, IQGAP1 knockdown experiments would be stronger if performed in iPSC neurons.

8. Fig. 4K-L – according to the data in Fig. 2B-C, NFAT luciferase activity reduction induced by INF- γ was the strongest in LRRK2 G2019S HEK cells; however in Fig. 4K-L, the INF- γ -induced reduction of NFAT luciferase in LRRK2 G2019S scrb HEK cells is very mild and not stronger than in LRRK2 wt scrb HEK cells – please explain this discrepancy.

9. Fig. 5C-D – The NRG1 rescue effect was demonstrated only in one LRRK2 PD and one isogenic CTRL iPSC lines. Given high variability, this important result needs a confirmation in additional two available iPSC lines (LRRK2 PD vs. CTRL).

INF- γ treatment in LRRK2 G2019S PD neurons did not lead to significant reduction of neurite length in Fig. 5D, which is in contrast to the previous data for this line (G2019S-2) shown in Fig. 5B. What is this difference occurring?

Minor points:

- It could be helpful for the readers to more precisely explain the "NFAT reporter HEK293 cell line": is Luciferase gene in this cell line expressed under a promoter with NFAT-response elements?
- How exactly knockdown of IQGAP1 in HEK cells was performed? Only lentiviral transduction is mentioned in the Mat&Meth.
- In the Abstract, mentioning which patients (PD?) are investigated is important.
- Introduction, the sentence about LRRK2 G2019S mutation as "the most common genetic cause of familial and sporadic PD" is not correct in terms of sporadic PD, where LRRK2 gene polymorphisms are not the strongest associated with PD risk.
- Introduction, the sentence stating that LRRK2 mutation is "most common pathogenic missense mutation" needs a specification that it is a "PD-associated" most common....
- Not all abbreviations are spelled out at the first use.

Reviewer #2 (Remarks to the Author):

The study of De Cicco et al, describes a linkage between the Parkinson's disease (PD) associated LRRK2 and IFN- γ using neuronal cultures and microglia differentiated from patients carrying the most common LRRK2 pathogenic mutation (G2019S) iPSCs. The study has several interesting observations assigning a novel function of LRRK2 as a linker between inflammation signaling pathways and

neurodegeneration. Although those observations are novel and of potential high impact they are not clearly connected and merged. There is an absence of cause effect relationship experiments. Another general observation which hinders the enthusiasm is that although the authors have generated TH neurons from PD patients and isogenic controls (i.e. they have generated a model with max disease relevance) they employ critical experiments in HEK 293 cells and non defined neuronal populations using β III tubulin. Although it is totally understood that there are technical challenges associated with the use of TH positive neurons, more key experiments need to be done in TH neurons. The use of statistics is not always clear. n=3 independent experiments for example it is not clear if it refers to different pairs of patients and isogenic controls or just replication of the same samples. The use of a two way ANOVA is recommended when there is treatment combined with genetic manipulation. (Figure 5 for instance). Also why SD and not SEM?

Specific comments

1. The increase in LRRK2 protein levels should be quantified in Figure 1
 2. It is not immediately clear why an increase of AKT3 in the mRNA level led the authors to test AKT phosphorylation. In fact in the absence of specific AKT3 phospho antibody they used phospho-pan AKT. Also the conclusion in page 6 "Taken together, these data indicate that IFN- γ leads to NFAT degradation by the proteasome and that the presence of the mutation G2019S enhances the retention of NFAT in the cytosol by reducing AKT phosphorylation" does not seem to be supported experimentally.
 3. In Figure 2A the expression of G2019S seems higher from WT and this could lead to altered behavior of the mutant expressing HEK293 cells in regards to NFAT activity. The use of a LRRK2 kinase inhibitor to reverse the findings would strengthen their conclusions
 4. In Figure 2D some of the neurons they choose to present do not seem healthy. For example the cells in control2 do not have a typical neuronal appearance and it is possible that this dramatic differences in the neuronal morphology could even affect neuronal localisation of NFAT quantification. A GFP plasmid either alone or encoding for another protein which translocates to the nucleus like NFAT can possibly address that. Also a detailed description of how the quantification of total vs nuclear fluorescence intensity is performed is missing.
 5. Calcium data is interesting but the direct contribution of NFAT/AKT/LRRK2 axis is missing.
 6. The microtubule data are very detailed and well performed albeit in HEK 293 cells. The effect of LRRK2 in microtubule dynamics is well established and is not novel. The authors have to elaborate different LRRK2 mediated signaling pathways previously reported to direct microtubule dynamics with their findings.
 7. The reversal of IFN γ mediated neurite outgrowth defect by LRRK2 kinase inhibitor is not restricted to the G2019S mice, indicating that LRRK2 expression and not the increased kinase activity conferred by the pathogenic mutation underlies the phenotype. The use of a more potent and specific inhibitor such as Mli-2 could yield cleaner results.
 8. In neurons NFAT activity is reduced in G2019S neurons at basal level, in microglia only after ionomycin induction. What does this mean? is this cell type specific? If this the case this highlights even more the importance to perform the studies in TH neurons.
- The microglia- neurons interaction experiment is well designed and pretty informative and a link between inflammation and neuronal functions
- A minor suggestion is to use the order of control and G2019S in the Figure presentation consistent throughout the manuscript, it is quite confusing.

Reviewer #3 (Remarks to the Author):

This manuscript is a nice mechanistic study demonstrating how IFN- γ and LRRK2 impact signaling pathways implicated in inflammation. Inflammation is clearly involved in Parkinson's disease pathogenesis and many studies show a role for LRRK2 in inflammation pathways. This study delves into the mechanisms by which LRRK2 acts downstream of IFN γ activation to inhibit NFAT nuclear

translocation in neuronal and microglial-like cells differentiated from iPSCs. One of the particularly nice aspect of the the study is the utilization of iPSCs derived from a Parkinson's patient with a G2019SLRRK2 mutation and isogenic controls. However, I do have a couple of concerns with the study:

In Figure 1 E-H: Why do the authors show mRNA fold changes over untreated for the control lines but not for the G2019S lines. The data for the G2019S is presented as change relative to control. I would like to see the data presented the same way for both control and G2019S. For example, mRNA fold changes over untreated in AKT3, STAT1, JaKC2, EIF2AK2, IFNG, IFNGR1 etc. in control and G2019S. It appears that some transcripts are not responsive to IFNgamma in G2019S cell lines compared to controls but the way the data is presented, makes this conclusion difficult.

In Figure 2C, it cannot be concluded that the most potent effect on NFAT was in G2019S because the authors did not test for statistical significance between the controls and G2019S. Furthermore, the t-test is not appropriate here- a 2 factor ANOVA with appropriate posthoc is the appropriate test.

In fact, throughout the paper the authors go between ANOVA and t-test, but in most cases t-test is not the appropriate statistic- such as figure 2C, 2E, 2G, 5B, 6D, 6E, 6F, 6G, 6H. I highly recommend the authors consult with a biostatistician.

In figure 3B, why do the authors present each control vs. G2019S group separately. Why don't the authors normalize the data, determine the averages for each of the 3 experimental groups, and then perform the statistics? In some figures, the authors do average the data of the 3 individual experiments and in some figures, they present the data separately. This does not seem to make sense.

In figure 4 B, the superresolution images are beautiful and there is clearly a morphological change in the microtubule structure in the G2019S LRRK2 cells. However, the authors cannot conclude that LRRK2 regulates NFAT shuttling through microtubules. To claim this the authors would have to rescue the microtubule curvature in the G2019S cells and determine if it restores nuclear translocation of NFAT in response to IFNgamma.

How do the authors reconcile the decreases in cytokine release in INFgamma treated G2019S LRRK2 cells (IL-6, IL-8, TNF-alpha) with the findings that these cytokines are increased in G2019S LRRK carriers? (see Dzamko et al movement disorders 31:889-897 2016). I also do not understand how if LRRK2 kinase activity prevents NFAT translocation to the nucleus, why is there an increase in cytokine release? Could another signaling pathway such as NFkappaB be involved here?

Are any of the phenotypes rescued by treatment with a LRRK2 kinase inhibitor. Does LRRK2 kinase inhibitor treatment increase shuttling of NFAT to the nucleus?

At the end of the manuscript it would be very helpful to have a figure summarizing the results of the signaling pathways in response to IFNgamma in control and G2019S cells. As of now, it is hard to understand the "takehome" message of this study.

Reviewer #4 (Remarks to the Author):

The authors show that interferon gamma (INF γ) in iPSC-derived neurons upregulates LRRK2 levels as well as other INF γ -regulated genes such as AKT3. Although increasing AKT3 mRNA and protein levels, INF γ decreases AKT3 phosphorylation at serine 473 and this phosphorylation is further reduced in iPSC neurons expressing LRRK2-G2019S. They confirmed previous finding that LRRK2 is a negative regulator of NFAT and showed LRRK2-G2019S further inhibits transcriptional activity of NFAT in HEK

cells. In iPSC-derived neurons, INF γ on LRRK2-G2019S appears to reduce nuclear localization of NFAT, in the absence of LRRK2, INF γ has no effect on NFAT nuclear localization. They further show alterations in intracellular calcium levels and microtubule dynamics in iPSC LRRK2-G2019S-derived neurons. Decrease in neurite length was found in control iPSC cells treated with INF γ , further decreases were observed in iPSC- LRRK2-G2019S-derived neurons and these latter neurons treated with INF γ . The authors have also evaluated cytokine profile and metabolic changes in iPSC or iPSC LRRK2-G2019S cells differentiated into microglia and showed condition medium from LRRK2-G2019S microglia cells reduces neurite length.

The approach of LRRK2 patient-derived cells and isogenic controls to examine the consequences of INF γ is relevant for pathophysiology of PD, however the following points need clarification and/or addressing:

1/ It is not clear in iPSC-derived neuronal cultures, the % of cells that have differentiated into dopaminergic neurons. It appears from images of Hoechst/beta tubulin staining that there may be non-neuronal cells; there is upregulation of ICAM1 by INF γ , ICAM1 is expressed in astrocytes but not known to be present in dopaminergic neurons.

2/In Fig. 1C the authors have used LPS to examine the specificity of INF γ action on LRRK2 - it's very likely that iPSC dopaminergic neurons do not express TLR4.

3/ Fig 2B and C: In Fig 2B, NFAT luciferase activity is greatly reduced in cells expressing LRRK2 or LRRK2-G2019S in the presence of PMA and ionomycin compared to control condition, however in Fig 2C these values are brought to 1 and reduction by INF γ ascertain what is the actual reduction in NFAT activity by INF γ in the presence of LRRK2.

In Fig. 2D images showing NFAT staining in nucleus are not convincing - staining appears cytoplasmic or perinuclear - this compared to clear nuclear staining in LRRK2 Ko condition (Fig. 2G). Would PMA/ionomycin treatment enhance nuclear translocation? INF γ treatment appears to be toxic to cells particularly in LRRK2-G2019S condition (Fig. 2D).

4/ Fig 3: Baseline intracellular Ca²⁺ levels in LRRK2-G2019S neurons are high suggesting Ca²⁺ buffering impairment and the Ca²⁺ influx in LRRK2-G2019S neurons following KCl treatment is lower - these result are opposite to those reported by Korecka et al 2018 - is there an explanation?

5/ The authors have observed neurite shortening by LRRK2-G2019S and INF γ that is reversed by neuregulin - does neuregulin correct abnormal Ca²⁺ levels and ER buffering capacity in these cells, Korecka et al 2018 showed that neurite collapse occurs as ER Ca²⁺ pump is inhibited. Are dopaminergic neurons particularly vulnerable to these effects of LRRK2-G2019S and INF γ ?

"Interferon- γ signaling synergizes with LRRK2 in human neurons and microglia "

General response to the reviewers

We would like to thank the reviewers for their insightful and detailed comments, which helped us substantially improve our manuscript. Their concerns and suggestions have been taken into careful consideration and we have made several additions and corrections to our work. We have fully addressed the issues raised by the Reviewers following multiple strategies to investigate the pathway(s) affected by mutant *LRRK2* in diseased neurons and microglia. In our view, with this new series of experiments, the novelty and the strength of our work has been considerably improved.

New data added in the revised version of the manuscript:

- In order to avoid artifacts related to the overexpression systems and provide data more relevant to the human disease condition, we have performed additional sets of experiments to confirm all the data initially obtained in HEK cells in human iPSC-derived neurons.
- A new series of experiments based on cytosolic/nuclear fractionation to validate the impact of LRRK2 and IFN- γ on NFAT nuclear translocation (Figure 2).
- A new series of experiments based on dSTORM imaging of iPSC-derived neurons aimed at characterizing the microtubule network in LRRK2 G2019S neurons (Figure 4).
- Functional experiments with stabilizing and destabilizing microtubule agents to demonstrate the role of the microtubule network in NFAT shuttling (Supplementary Figure 3).
- Ca²⁺ imaging experiments addressing the mechanisms by which neuregulin 1 rescues the impairment of NFAT activation and neurite growth in iPSC-derived neurons (Supplementary Figure 5).
- Further characterization of the described phenotypes in iPSC-derived dopaminergic neurons (Supplementary Figure 4,5, and 10).
- Characterization of the impact of an additional LRRK2 inhibitor (MLi-2) in iPSC-derived neurons (Figure 4 and Supplementary Figure 4).
- Improved characterization of the efficiency of iPSC-derived DA neurons (Supplementary Figure 1) and microglia differentiation (Supplementary Figure 6).
- Improved functional characterization of control and patient microglial cells, including microglial phagocytosis, motility, energy metabolism (Figure 6), and analysis of NF- κ B signaling pathway (Supplementary Figure 8).

- Assessment of the role of LRRK2 kinase activity in the regulation of microglial activation (Figure 6 and Supplementary Figure 9).

#Reviewer 1

- **Q1: Fig. 1 – How do the used IFN- γ concentrations correlate with physiological concentration measured in serum or CSF samples of PD patients?**

While most of the *in vitro* studies employ 100 ng/ml IFN- γ , in the present study we have used a rather low concentration of IFN- γ , namely 200 IU/ml, which equals 10ng/ml. The concentration of IFN- γ in human serum and CSF is quite low (\approx 15pg/ml and undetectable, respectively). These levels have been found to be increased in PD patients' serum (Brodacki et al., 2008). In PD patients' substantia nigra the concentration of IFN- γ was found \approx 300 pg/mg of protein. This is significantly higher compared to healthy subjects (\approx 170 pg/mg) and this increase was specific to substantia nigra (IFN- γ levels were not increased in other brain regions) ¹. However, one cannot directly compare cytokine concentrations between *in vivo* and *in vitro* systems. As a matter of fact, *in vitro* cell culture is a substantially different environment that cannot thoroughly resemble the complexity of the human brain. The effect of an individual cytokine *in vivo* is modulated by the presence or absence of other cell types, cytokines, growth factors etc.

- **Q2: Is IFN- γ -mediated LRRK2 increase specific to DA neurons? Why is the dose explicitly mentioned for the second paragraph of results but not for the first?**

Thank you for this interesting comment; to address this question we investigated the effect of IFN- γ treatment on LRRK2 levels in cortical neurons. Human iPSCs were differentiated into cortical neurons as described in Perez, Ivanyuk et al. ² and treated with 200IU/mL IFN- γ for 24hrs. Our results revealed that the IFN- γ -mediated induction of LRRK2 also occurs in cortical neuronal cultures (Supplementary Fig. 1D). However, being differentiated cultures enriched in, and not pure, DA and cortical neuronal cultures, one cannot exclude a cell-type specific regulation of LRRK2. We have now explicitly mentioned the IFN- γ dose (which is 200 IU/ml in all the experiments except form the initial dose-range experiment shown in Figure 1).

- **Q3: Fig. 2 – The level of overexpression of LRRK2 G2019S seems to be slightly higher in HEK cells than the overexpression of LRRK2 WT in Fig. 2A. Consequently, only slightly lower levels of NFAT luciferase after PMA/Ionomycin stimulation were determined in HEK cells overexpressing LRRK2 G2019S compared to LRRK2 WT HEK cells in Fig. 2B. Thus, it might be not an effect of the mutant LRRK2 form, but rather an effect of higher**

LRRK2 protein. Moreover, if the PMA/Ion-mediated NFAT luciferase increase in LRRK2 G2019S HEK cells is not significantly different from the basic line (unstimulated HEK cells; Fig. 2B), how effective any other treatment including IFN- γ treatment may be in reducing its activity? Maybe combining data on Fig. 2B and 2C would give more clear picture to the real reduction rate in different HEK cells by IFN- γ .

Thank you for highlighting these points. LRRK2 levels were comparable in HEK cells overexpressing G2019S compared to wild type, as revealed upon densitometric quantification (1.03 in LRRK2 G2019S overexpressing cells normalized to LRRK2 wt). In Supplementary Figure 3 (revised version) we provide additional images that more faithfully represent the actual, comparable, levels of LRRK2. We would also like to highlight that, to avoid artifacts related to the overexpression systems and provide data more relevant to the human disease condition, we have performed additional sets of experiments to confirm all the data initially originated in HEK cells in human iPSC-derived neurons. More specifically, this part has now been replaced by data showing NFAT translocation using nuclear/cytosolic fractionation and subsequent Western blot analysis (Fig. 2 and Supplementary Fig. 1).

- ***Q4: it is hard to detect any NFAT nuclear localization in neurons on the provided pictures (Fig. 2D); Would neurons need any additional stimulation for NFAT nuclear translocation similar to what was done for HEK cells or for neurons in the Suppl. Fig. 1H (Ionomycin treatment)? What was the reason for NFAT3 overexpression in iPSC-derived neurons? Instead, an evaluation of endogenous NFAT3 by means of, for example, immunofluorescence staining would firstly, stress a relevance of NFAT3 in neurons; and secondly, provide a physiological influence of IFN- γ on NFAT3 neuronal localization.***

We agree with the reviewer that endogenous levels as assessed by immunofluorescent staining would be more relevant. We have tested several commercial antibodies; however, the lack of specific signal for NFAT3 prevented us from employing them for such analysis. Due to the lack of validated anti-human NFAT3 antibodies, we have initially employed overexpression of an NFAT-GFP reporter. We certainly agree that the overexpression systems and immunofluorescent quantification in iPSC-derived neurons may present some limitations. For this reason, along with the biochemical data initially provided in Supplementary Fig. 1, we have now performed an additional set of experiments related to the quantification of endogenous NFAT localization using nuclear and cytosolic fractionation techniques. These biochemical data further support decreased NFAT nuclear shuttling in LRRK2 G201S iPSC-derived neurons and the negative impact of IFN- γ . In the revised version of the manuscript, we also provide data

related to ionomycin-induced NFAT activation. These additional data are provided in Fig. 2 and Supplementary Fig. 1. Furthermore, we also provide data relative to the levels of phospho-NFAT3, which is the cytosolic, inactive form. Increased levels of phospho-NFAT3 were found in LRRK2 G2019S neurons. These additional data, which are now provided Supplementary Fig. 3, further strengthen the negative impact of LRRK2 G2019S on NFAT activity.

Furthermore, we now provide data relative to ionomycin stimulation (Figure 2, Supplementary Figure 1,2, and 3). Our data show that the decrease of NFAT nuclear localization is already present in G2019S neurons at basal level, whereas in HEK cells (and microglia) is induced upon ionomycin stimulation. Furthermore, it is interesting to note that ionomycin-dependent stimulation of NFAT was comparable in control and G2019S neurons. However, it is well known that the cytosolic Ca^{2+} overloading induced by ionomycin strongly stimulates Ca^{2+} influx in neurons, leading to the activation of several other cellular pathways including cell death pathways. For this reason, a short-term treatment with a low dose of ionomycin was employed in our experiments more as a positive control, namely to show the activation of NFAT in human iPSC neurons in response to increase of intracellular Ca^{2+} . In general, our findings support previous literature showing that NFAT activation is cell-type specific mostly depending on calcium dynamics. For example, it is known that specialized L-type Ca^{2+} channels regulate NFAT3 translocation and function in hippocampal neurons ³. More recent work has also highlighted a role for the synapse-nucleus signaling in the activity-dependent transcription factor NFAT, whereby somatic propagation of LTCC- Ca^{2+} spikes initiated in dendrites trigger translocation of NFAT ⁴. It is also interesting to note that Ca^{2+} signaling differentially regulate key effector functions in adaptive and innate immune cells (macrophages and dendritic cells) ⁵. We would like to thank the Reviewer for highlighting this important observation.

- **Q5: The relevance of NFAT in neurons needs to be mentioned at this point (it is mentioned first much later in the pre-last part of the Results).**

Thank you for this suggestion. Appropriate modifications have been made to the manuscript: Page 6: "*NFAT family members are regulators of Ca^{2+} -dependent gene transcription in a variety of cell types, including immune cells and neurons ⁶. In the brain, NFAT transcription complexes play important functions in neuronal growth and brain circuit formation ⁶*"

- **Q6: What individual dots in E, F, G represent? If these are individual experiments (independent differentiation rounds?) as stated in the figure legend, that please explain**

how many cells were analyzed in each experiment.

Exactly, in previously submitted Fig. 2, each dot represented the average value from 5-10 cells of one experiment; each experiment consists of a separate differentiation round. However, as mentioned above, in the revised version of the manuscript these graphs have been replaced with data showing NFAT translocation using nuclear fractionation experiments in human iPSC-derived neurons.

- ***Q7: LRRK2 KO iPSC lines (Suppl. Fig. 1E-F) are raising several questions: LRRK2 bands in left and right parts of the figure corresponding to CTRL-1 and CTRL-4, respectively, do have a different pattern and this needs to be explained.***
- ***LRRK2 bands are different in CTRL-1+/+ samples on the left and right CTRL-1 WB pictures (and also different in the CTRL-4+/+ samples on the left and right CTRL-4 WB pictures): these differences must be explained;***
- ***More detailed labeling of the Suppl. Fig. 1E could be helpful to better understand the figure message;***

We understand the reviewer's concern and we agree that we have not provided sufficient information. Since different strategies were employed to generate the LRRK2 KO clones from two different parental lines, we assesses the lack of LRRK2 protein using different antibodies:

- 1E11
- 24D8

previously developed by Dr Elisabeth Kremmer, Helmholtz Zentrum München, Munich, Germany^{7,8}

- UDD3 antibody, LRRK2 aa 100-500 (from Abcam)
- MJFF2, [MJFF2 (c41-2)] (from Abcam)

Hence, a different pattern was detected. Original Western blot images are provided in the data source file. As suggested, we now provide additional information in the figure labeling and supplementary figure legend (Supplementary Fig. 2A).

- ***Q8: LRRK2 protein signal in +/+ samples in the Suppl. Fig. 1E looks slightly different from the LRRK2 signal in Suppl. Fig. 1B: does this represent differences between iPSC-derived neurons (1B) and microglial cells (1E)? Please explain;***

Yes and also due to the fact that in Supplementary Fig. 1B we employed a different antibody.

- **Q9: LRRK2 KO iPSC-derived neurons or microglia cells must be characterized for at least neuronal and microglial markers in order to assess differentiation efficiency.**

Thank you for this comment. Both neurons and microglia have been characterized for the differentiation efficiency and these data have been included in the revised manuscript (Supplementary Fig. 2 and Fig. 6). Furthermore, we have now provided additional characterization of microglia function including phagocytic capacity, motility, and energy metabolism (Fig. 6).

- **Q10. iPSC-derived microglia cells require more detailed characterization, which is completely omitted in the manuscript. First results from iPSC-derived microglial cells are presented “suddenly” in the Suppl. Fig. 1D-E without providing a clear rationale for using these cells to evaluate NFAT and LRRK2 expression and without any characterization of the generated microglia cells. Moreover, taking into account that total cell lysates were used to evaluate NFAT mRNA (Suppl. Fig. 1D) and LRRK2 protein (Suppl. Fig. 1E) expression, without knowing the efficiency of microglial cell differentiation and the frequency of microglial cells in the resulting cultures, it is hard to attribute respective signals to microglial cells.**

We agree and a detailed microglial characterization, including immunofluorescent staining with specific markers and assessment of phagocytic capacity, has been included in the revised manuscript (Fig. 6 A-E).

- **Q11. Why in Suppl. Fig. 1F WB picture (on the left) corresponds to NFAT3 and the quantification (on the right) is performed on total NFAT? Please clarify or show total NFAT WB signals, which were quantified. Please clarify discrepancy between the legend to the Suppl. Fig. 1F-I stating “total NFAT3” and the text of the Results part (page 6) saying “total NFAT”.**

Thank you for pointing this out. The term “total NFAT” was referring to total NFAT3 levels. In the revised version of the manuscript, we have clarified the difference between “total NFAT3” (nuclear and cytosolic fraction) and “NFAT3 from whole-cell extracts”.

- **Q12. NFAT3 signal in Suppl. Fig. 1G-H looks different from NFAT3 signal in Suppl. Fig. 1F and 1I. Why NFAT3 signal looks different between total cell lysates (1F, 1I) and in cytosolic and nuclear fractions (1G-H).**

The reviewer is correct, the signal is different. In Suppl. Fig. 1F and 1I (former version) we showed NFAT3 from whole cell extracts, whereas in Suppl. Fig. 1G-H cellular fractionation Western blots were shown. We have now better clarified this issue.

- **Q13. Why in fractionation experiments presented in Suppl. Fig. 1H an additional Ionomycin stimulation was performed and in 1G not?**

In the revised version of the manuscript we have now included data related to the fractionation experiments with and without ionomycin stimulation and IFN- γ treatment (Fig. 1 and Supplementary Fig. 1). As detailed in our answer to Q4, a short-term treatment with a low dose of ionomycin was employed in our experiments more as a positive control, namely to show the activation of NFAT in human iPSC neurons in response to increase of intracellular Ca^{2+} .

- **Q14: Fig. 4A-I – the results on microtubule network differences depending on LRRK2 mutation are very interesting, but shown only in HEK cells. The relevance of this phenomenon to human PD pathology could be strengthened by evaluation of microtubule state (stable vs. unstable) or microtubule networks in iPSC-derived neurons from LRRK2 G2019S PD patients compared to isogenic controls.**

Thank you for this suggestion. It is known that microtubule acetylation increases the stability of their structure. Thus, in order to address this point, we have performed Western blot analysis on acetylated and non-acetylated tubulin levels in LRRK2 PD and isogenic control iPSC-derived neurons. However, no significant differences were observed (Figure 1). Furthermore, as suggested, we have performed an entire new set of dSTORM imaging to evaluate the microtubule network in iPSC-derived neurons. These data have now entirely replaced the previous dataset obtained in HEK cells and they are now provided in Fig. 4 in the revised version of the manuscript.

Figure 1. Analysis of acetylated tubulin in LRRK2 G2019S and isogenic control neurons.

- **Q15. Moreover, IQGAP1 knockdown experiments would be stronger if performed in iPSC neurons.**

This is an interesting point. To strengthen the proposed IQGAP-mediated mechanism, we have performed siRNA-mediated knockdown of IQGAP1 (Silencer® Select, ThermoFisher) in both LRRK2-PD and control iPSC-derived neurons. While we could achieve a high efficiency in silencing, IQGAP1 knockdown led to disruption of the neuritic network in both LRRK2-PD and isogenic controls neurons (Figure 2). These data are in agreement with the well-known role of IQGAP1 in neurite outgrowth and control of dendrite morphology^{9,10}. Thus, it is challenging to dissect IQGAP1-dependent mechanisms from the phenotypes linked to a general disruption of the cytoskeleton and microtubule network. Specifically, the disruption of the neuritic network may result in a confounding dysregulation of NAFT translocation. Due to these findings, we are unable to provide solid data in human iPSC-derived neurons. Based on the fact that the entire manuscript is now based on data generated from patient cells, we have decided to remove the IQGAP1 data from the revised version.

Figure 2. IQGAP1 knockdown in human iPSC-derived neurons.

- **Q16. Fig. 4K-L – according to the data in Fig. 2B-C, NFAT luciferase activity reduction induced by INF-γ was the strongest in LRRK2 G2019S HEK cells; however in Fig. 4K-L, the INF-γ-induced reduction of NFAT luciferase in LRRK2 G2019S scrb HEK cells is very mild and not stronger than in LRRK2 wt scrb HEK cells – please explain this discrepancy.**

We agree with the reviewer's concern. There was a mistake in the normalization of the data in

Fig. 4 in the previous version of our manuscript. We apologize for this, the correct graph should appear as follows:

However, the Reviewer's observation is correct, we also noticed a reduction in the response to INF-γ treatment (as well as to ionomycin) in the NFAT HEK reporter cell lines we have employed for IQGAP1 experiments. We hypothesize that the lentiviral-mediated delivery of shRNA and scrb vectors employed in these experiments may interfere with and attenuate the response to a subsequent immunological stimulus. Indeed, it is known that lentiviral transduction activates intracellular antiviral defense mechanisms. Based on these premises and the fact that it is not feasible to provide IQGAP1-related data in human neurons, due to the toxicity of the knockdown, we have removed these data from the revised version of the manuscript.

- **Q17. The NRG1 rescue effect was demonstrated only in one LRRK2 PD and one isogenic CTRL iPSC lines. Given high variability, this important result needs a confirmation in additional two available iPSC lines (LRRK2 PD vs. CTRL).**

We agree and we have now confirmed the rescue effect of NRG1 in two additional available iPSC lines (LRRK2 PD vs. CTRL) (Fig. 5). Furthermore, quantification of TH-positive neurites is now provided in Supplementary Fig 5.

- **Q19. INF-γ treatment in LRRK2 G2019S PD neurons did not lead to significant reduction of neurite length in Fig. 5D, which is in contrast to the previous data for this line (G2019S-2) shown in Fig. 5B. What is this difference occurring?**

We indeed found a reduction of neurite length upon INF-γ treatment. However, in this set of experiments it did not reach significance. As suggested in Q18, we have now performed

additional experiment to include all the isogenic pairs (Fig. 5).

Minor points:

- ***It could be helpful for the readers to more precisely explain the “NFAT reporter HEK293 cell line”: is Luciferase gene in this cell line expressed under a promoter with NFAT-response elements?***

Since the lines are commercial, we did not include further additional technical details. However, we do agree and we have now included additional information as follows: "These cells contain a luciferase gene under the control of Nuclear Factor of Activated T-cell response elements stably integrated into HEK293 cells"

- ***How exactly knockdown of IQGAP1 in HEK cells was performed? Only lentiviral transduction is mentioned in the Mat&Meth.***

Information was provided on page 17 under "Luciferase experiments".

- ***In the Abstract, mentioning which patients (PD?) are investigated is important.***

Thank you, we apologize if this was not clearly mentioned. We have now rephrased this part to clearly state that we have investigated LRRK2 (G2019S) PD patients.

- ***Introduction, the sentence about LRRK2 G2019S mutation as “the most common genetic cause of familial and sporadic PD” is not correct in terms of sporadic PD, where LRRK2 gene polymorphisms are not the strongest associated with PD risk.***

We agree and the correction has been made.

- ***Introduction, the sentence stating that LRRK2 mutation is “most common pathogenic missense mutation” needs a specification that it is a “PD-associated” most common....***

Thank you for pointing this out, it has been modified.

- ***Not all abbreviations are spelled out at the first use.***

We apologize for these errors, which have been corrected in the revised manuscript.

#Reviewer 2

The study of De Cicco et al, describes a linkage between the Parkinson's disease (PD) associated LRRK2 and IFN- γ using neuronal cultures and microglia differentiated from patients carrying the most common LRRK2 pathogenic mutation (G2019S) iPSCs. The study has several interesting observations assigning a novel function of LRRK2 as a linker between inflammation signaling pathways and neurodegeneration. Although those observations are novel and of potential high impact they are not clearly connected and merged. There is an absence of cause effect relationship experiments. Another general observation which hinders the enthusiasm is that although the authors have generated TH neurons from PD patients and isogenic controls (i.e. they have generated a model with max disease relevance) they employ critical experiments in HEK 293 cells and non-defined neuronal populations using β III tubulin.

Although it is totally understood that there are technical challenges associated with the use of TH positive neurons, more key experiments need to be done in TH neurons.

The use of statistics is not always clear. n=3 independent experiments for example it is not clear if it refers to different pairs of patients and isogenic controls or just replication of the same samples. The use of a two-way ANOVA is recommended when there is treatment combined with genetic manipulation. (Figure 5 for instance). Also, why SD and not SEM?

We thank the Reviewer for acknowledging the importance and the novelty of our findings. We acknowledge previous limitations and agree that the analysis of the functional consequences of LRRK2 mutations in iPSC-derived DA neurons was missing in the original version of our manuscript. We have therefore fully addressed these issues by following multiple strategies to investigate the pathway(s) affected by mutant *LRRK2*. As described below, as well as in our response to other Reviewers, we looked in more detail and clarified the mechanisms of NFAT activation in diseased neurons. We also agree that data obtained in disease relevant cells would provide more valuable information compared to those generated in immortalized cell lines using overexpression systems. For this reason, we have performed new sets of experiments of d-STORM imaging and nuclear/cytosolic fractionation in human iPSC-derived neurons and functional experiments to assess the role of microtubules in NFAT activation in LRRK2 cells. Furthermore, we have performed key experiments in TH-positive neurons. Finally, we have

adjusted statistics where appropriate. In our view, with this new series of experiments, we have considerably improved the novelty and the strength of our work.

- **Q1. The increase in LRRK2 protein levels should be quantified in Figure 1**

Thank you, the quantification has now been included (Fig. 1B).

- **Q2. It is not immediately clear why an increase of AKT3 in the mRNA level led the authors to test AKT phosphorylation. In fact in the absence of specific AKT3 phospho antibody they used phospho-pan AKT. Also the conclusion in page 6 "Taken together, these data indicate that IFN- γ leads to NFAT degradation by the proteasome and that the presence of the mutation G2019S enhances the retention of NFAT in the cytosol by reducing AKT phosphorylation" does not seem to be supported experimentally.**

We agree with the Reviewer that the reasoning led us to the analysis of phospho-AKT was not clearly described in our previous version and we have now better clarified this point. Concerning the role of AKT3, we have shown that IFN- γ suppresses AKT3 phosphorylation -hence, activation- in human iPSC-derived neurons. Normally, AKT inactivates GSK3 β , which phosphorylates NFAT leading to its inactivation and cytosolic shuttling. Hence, our data suggest that IFN- γ negatively regulates the nuclear shuttling of NFAT, at least in part, by the AKT/ GSK3 β pathway. Indeed, LRRK2 kinase activity is associated to increased GSK3 β activation in a variety of model systems including human iPSC-derived neurons^{11, 12, 13, 14}.

As mentioned in our manuscript, due to the lack of phospho-AKT3-specific antibodies for Western blot, an antibody recognizing all AKT isoforms was initially used. During the revision process, we have explored different approaches for phospho-Akt3 quantification, including testing additional antibodies. We have therefore proceeded to specifically quantify phospho- (and total-Akt3 levels), using the Milliplex Magnetic Bead Panel (EMD Millipore Catalog # 48-632MAG). These new experiments have confirmed the initial Western blot data, obtained using total and phospho-AKT (pan) antibodies.

- **Q3. In Figure 2A the expression of G2019S seems higher from WT and this could lead to altered behavior of the mutant expressing HEK293 cells in regard to NFAT activity. The use of a LRRK2 kinase inhibitor to reverse the findings would strengthen their conclusions.**

LRRK2 levels were comparable in HEK cells overexpressing G2019S compared to wild type, as revealed upon densitometric quantification (1.03 in G2019S overexpressing cells normalized to wt). In Supplementary Figure 3 in the revised version we provide additional images that more faithfully represent the actual, comparable, levels of LRRK2. To avoid artifacts related to the overexpression systems and provide data more relevant to the human disease condition, we have performed additional sets of experiments to confirm all the data initially originated in HEK cells in human iPSC-derived neurons. More specifically, this part has now been replaced by data showing NFAT translocation using nuclear/cytosolic fractionation and subsequent Western blot analysis (Fig. 2 and Supplementary Fig. 1).

- ***Q4. In Figure 2D some of the neurons they choose to present do not seem healthy. For example the cells in control2 do not have a typical neuronal appearance and it is possible that this dramatic differences in the neuronal morphology could even affect neuronal localization of NFAT quantification. A GFP plasmid either alone or encoding for another protein, which translocates to the nucleus like NFAT can possibly address that. Also a detailed description of how the quantification of total vs nuclear fluorescence intensity is performed is missing.***

We agree with the limitations of our approach based on the overexpression of NFAT-GFP. As pointed out by Reviewer 1, data related to endogenous levels would be more relevant. We have tested several commercial antibodies; however, the lack of specific signal prevented us from employing them for such analysis. Due to the lack of validated anti-human NFAT3 antibodies, we have initially employed overexpression of an NFAT-GFP reporter. We do agree that the overexpression systems and immunofluorescent quantification in iPSC-derived neurons may present some limitations. For this reason, along with the biochemical data initially provided in Supplementary Fig. 1, we have now performed an additional sets of experiments related to the quantification of endogenous NFAT localization by nuclear and cytosolic fractionation experiments. These biochemical data further support decreased NFAT nuclear shuttling in LRRK2 mutant iPSC-derived neurons and the negative impact of IFN- γ . In the revised version of the manuscript, we now provide additional data related to the activation pathway of NFAT by ionomycin stimulation. These additional data that are now provided in Fig. 2 and Supplementary Fig. 1 further strengthen the link between LRRK2 and NFAT activity. Furthermore, we also provide data relative to the levels of phospho-NFAT3, which is the cytosolic, inactive form. Increased levels of phospho-NFAT3 were found in LRRK2 G2019S neurons. These additional data, which are now provided Supplementary Fig. 3, further strengthen the link between LRRK2

and NFAT activity.

- **Q5. Calcium data is interesting but the direct contribution of NFAT/AKT/LRRK2 axis is missing.**

We would like to point out that our work shows the contribution of calcium dynamics to the LRRK2/NFAT axis and not the direct contribution of the NFAT/AKT/LRRK2 axis to calcium dynamics. Since Ca^{2+} regulates NFAT activation, our data showing a defect in intracellular calcium buffering capacity and decreased calcium release along with the newly performed rescue experiments with NRG1, support the role of calcium dynamics in LRRK2-mediated regulation of NFAT shuttling. Specifically, we now provide data showing that NRG1, which activates NFAT³, rescues the impairment of ER Ca^{2+} buffering in LRRK2 G2019S on calcium dynamics and defects of neurite elongation induced by either LRRK2 or IFN- γ (Supplementary Fig. 5).

- **Q6. The microtubule data are very detailed and well performed albeit in HEK 293 cells. The effect of LRRK2 in microtubule dynamics is well established and is not novel. The authors have to elaborate different LRRK2 mediated signaling pathways previously reported to direct microtubule dynamics with their findings.**

Thank you for this comment. We agree and have therefore included a, more relevant, new set of microtubule network dSTORM imaging analysis in iPSC-derived CTRL and LRRK2 G2019S neurons. Interestingly, the differences in microtubule complexity previously shown in HEK 293 cells, are present, and even more enhanced, in neuronal cells.

These results are included in the revised version of our manuscript as a new Figure 4.

- **Q7. The reversal of IFN- γ mediated neurite outgrowth defect by LRRK2 kinase inhibitor is not restricted to the G2019S mice, indicating that LRRK2 expression and not the increased kinase activity conferred by the pathogenic mutation underlies the phenotype. The use of a more potent and specific inhibitor such as Mli-2 could yield cleaner results.**

We are not sure whether we fully understand this comment, since IFN- γ increases levels of LRRK2 (and thus kinase activity) in both control and LRRK2 G2019S neurons. Consequently, LRRK2 kinase inhibition also has a rescue effect on control iPSC-derived neurons upon IFN- γ treatment. We agree with the Reviewer that Mli-2 is the most potent and specific inhibitor, thus we have performed additional experiments and we now provide data related to this LRRK2

inhibitor. Thank you for this suggestion.

These results are included in the revised version of our manuscript in Figure 5 and Supplementary Figures 4-5.

- ***Q8. In neurons NFAT activity is reduced in G2019S neurons at basal level, in microglia only after ionomycin induction. What does this mean? is this cell type specific? If this the case this highlights even more the importance to perform the studies in TH neurons.***

The Reviewer is right; our data show that the decrease of NFAT nuclear localization is already present in G2019S neurons at basal level, whereas in microglia (and HEK cells) it is induced upon Ionomycin stimulation. This is a cell-type specific behaviour depending on cell-type specific calcium dynamics. For example, it is known that specialized L-type Ca^{2+} channels regulate NFAT3 translocation and function in hippocampal neurons ³. More recent work has also highlighted a role for the synapse-nucleus signaling in the activity-dependent transcription factor NFAT, whereby somatic propagation of LTCC- Ca^{2+} spikes initiated in dendrites trigger translocation of NFAT ⁴. It is also interesting to note that Ca^{2+} signaling differentially regulate key effector functions in adaptive and innate immune cells (macrophages and dendritic cells) ⁵. This evidence would support our findings showing that the key effector functions in microglia are regulated in an NFAT-independent fashion. We would like to thank the Reviewer for highlighting this important observation and we have now addressed this issue in the discussion.

Q9. The microglia- neurons interaction experiment is well designed and pretty informative and a link between inflammation and neuronal functions. A minor suggestion is to use the order of control and G2019S in the Figure presentation consistent throughout the manuscript, it is quite confusing.

Thank you. This correction has been done.

#Reviewer 3

Reviewer #3 (Remarks to the Author):

This manuscript is a nice mechanistic study demonstrating how IFN- γ and LRRK2 impact signaling pathways implicated in inflammation. Inflammation is clearly involved in Parkinson's disease pathogenesis and many studies show a role for LRRK2 in inflammation pathways. This study delves into the mechanisms by which LRRK2 acts downstream of IFN- γ activation to inhibit NFAT nuclear translocation in neuronal and

microglial-like cells differentiated from iPSCs. One of the particularly nice aspect of the study is the utilization of iPSCs derived from a Parkinson's patient with a G2019SLRRK2 mutation and isogenic controls. However, I do have a couple of concerns with the study: We thank the reviewer for the positive comments and we have now addressed the concerns as follows:

- **Q1. In Figure 1 E-H: Why do the authors show mRNA fold changes over untreated for the control lines but not for the G2019S lines. The data for the G2019S is presented as change relative to control. I would like to see the data presented the same way for both control and G2019S. For example, mRNA fold changes over untreated in AKT3, STAT1, JaKC2, EIF2AK2, IFNG, IFNGR1 etc. in control and G2019S. It appears that some transcripts are not responsive to IFN γ in G2019S cell lines compared to controls but the way the data is presented, makes this conclusion difficult.**

Thank you for pointing this out. The reason of showing mRNA fold changes over untreated for the control lines first was to test whether interferon type II responses is conserved in human neurons *in vitro*. Next, we investigated whether LRRK2 influences the response to IFN- γ treatment, thus our scope was to highlight the differences in expression levels between the control and mutant lines upon IFN- γ treatment. We have better clarified this point within the text. Nonetheless, mRNA fold changes over untreated for G2019S lines are now included in the source data file.

- **Q2. In Figure 2C, it cannot be concluded that the most potent effect on NFAT was in G2019S because the authors did not test for statistical significance between the controls and G2019S. Furthermore, the t-test is not appropriate here- a 2 factor ANOVA with appropriate posthoc is the appropriate test. In fact, throughout the paper the authors go between ANOVA and t-test, but in most cases t-test is not the appropriate statistic- such as figure 2C, 2E, 2G, 5B, 6D, 6E, 6F, 6G, 6H.**

We agree and we have now revised all our statistics.

- **Q3. In figure 3B, why do the authors present each control vs. G2019S group separately. Why don't the authors normalize the data, determine the averages for each of the 3 experimental groups, and then perform the statistics? In some figures, the authors do average the data of the 3 individual experiments and in some figures, they present the data separately. This does not seem to make sense.**

Thank you for this suggestion. We have now combined the data from the different isogenic couples for data representations.

- **Q4. In figure 4 B, the superresolution images are beautiful and there is clearly a morphological change in the microtubule structure in the G2019S LRRK2 cells. However, the authors cannot conclude that LRRK2 regulates NFAT shuttling through microtubules. To claim this the authors would have to rescue the microtubule curvature in the G2019S cells and determine if it restores nuclear translocation of NFAT in response to IFN γ .**

This is a very interesting point and we have been discussing thoroughly on how to address this issue. In general, while it is feasible to modulate the biophysical properties of individual microtubules (e.g. with MicroTweezers), these experiments cannot be performed on the entire microtubule network in intact cells. As an alternative, we have conducted experiments using NFAT reporter HEK cell lines that clearly show the effect of microtubule-stabilizing and destabilizing agents on NFAT translocation (Supplementary Fig. 3). Specifically, we found that treatment with colchicine, an inhibitor of microtubule polymerization, leads to impaired NFAT activation, whereas treatment with the microtubule stabilizer, Paclitaxel, is associated with an increase of NFAT translocation/activation. Interestingly, this positive effect was stronger in G2019S cells, suggesting a relevant role of LRRK2-associated changes of the microtubule network in NFAT shuttling. Furthermore, we have validated these findings in isogenic control and G2019S neurons. iPSC-derived neurons were treated with PTX and immunostained for phospho-NFAT3, the cytosolic, inactive, form of NFAT3. Basal levels of phospho-NFAT3 were significantly higher in G2019S compared to control neurons. PTX significantly decreased the levels of phospho-NFAT3, indicating increased NFAT3 nuclear translocation. These findings support the link between LRRK2-mediated changes in the microtubule network and impairment of NFAT shuttling.

These results are included in the revised version of our manuscript in Supplementary Figure 3.

- **Q5. How do the authors reconcile the decreases in cytokine release in INF γ treated G2019S LRRK2 cells (IL-6, IL-8, TNF- α) with the findings that these cytokines are increased in G2019S LRRK carriers? (see Dzamko et al movement disorders 31:889-897 2016). I also do not understand how if LRRK2 kinase activity prevents NFAT translocation to the nucleus, why is there an increase in cytokine release? Could another signaling pathway such as NF κ B be involved here?**

The Reviewer is correct. As stated in our discussion "additional mechanisms other than NFAT

regulate cytokine responses in LRRK2 KO and G2019S microglia". Despite the impairment of NFAT translocation, overall LRRK2 G2019S microglia present a more active phenotype. Many functional aspects of myeloid cell activation are NFAT-independent, including phagocytosis and cellular metabolism. We have also addressed potential mechanisms underlying the LRRK2-dependent regulation of the key effector functions of microglia. In the revised manuscript we show that LRRK2 G2019S impairs the nuclear translocation of the NF- κ B p65 subunit upon LPS and IFN- γ stimulation. Besides its well-known role in pro-inflammatory gene expression, NF- κ B has important anti-inflammatory and immune-suppressive functions. Furthermore, inhibition of NF- κ B has been associated to cellular metabolic reprogramming towards aerobic glycolysis¹⁵. Interestingly, this is in line with our results showing that LRRK2 regulates microglia activation by interfering with the metabolic switch toward glycolysis, which normally occurs in LPS- and IFN- γ -activated macrophages.

These results are included in the revised version of our manuscript in Figure 6 and Supplementary Figures 8-10.

- ***Q6. Are any of the phenotypes rescued by treatment with a LRRK2 kinase inhibitor? Does LRRK2 kinase inhibitor treatment increase shuttling of NFAT to the nucleus?***

Thank you for this suggestion. Since, as correctly pointed out by the reviewer, our data show that the key effector functions of microglia are regulated in an NFAT-independent manner, we have assessed the impact of LRRK2 kinase inhibition on cytokine production and metabolic reprogramming, rather than NFAT shuttling. Interestingly, for the majority of cytokines tested, LRRK2 kinase inhibition restores their levels towards the one observed in control lines (Supplementary Fig. 9). Furthermore, inhibiting LRRK2 kinase attenuates the glycolytic switch upon LPS and IFN- γ stimulation (Fig. 6). Taken together, these data suggest that LRRK2 kinase activity contribute to microglial homeostasis and activation.

- ***Q7. At the end of the manuscript it would be very helpful to have a figure summarizing the results of the signaling pathways in response to IFN γ in control and G2019S cells. As of now, it is hard to understand the "takehome" message of this study.***

Thank you for this suggestion, a graphical abstract is included in the revised version of the manuscript (Fig. 7).

Reviewer #4

We thank the reviewer for the suggestions that we have taken into consideration as follows.

- **Q1. It is not clear in iPSC-derived neuronal cultures, the % of cells that have differentiated into dopaminergic neurons. It appears from images of Hoechst/beta tubulin staining that there may be non-neuronal cells; there is upregulation of ICAM1 by INF γ , ICAM1 is expressed in astrocytes but not known to be present in dopaminergic neurons.**

We agree with the reviewer and we have now included quantification of neuronal differentiation in our cultures (Supplementary Fig. 1A, B). It is true that ICAM1 is mostly expressed in astrocytes and endothelial cells. However, ICAM1 is also known to be expressed on many other cell types, including neurons, and its expression is upregulated by proinflammatory cytokines, such as IFN- γ ^{16 17}.

- **Q2. In Fig. 1C the authors have used LPS to examine the specificity of INF γ action on LRRK2 - it's very likely that iPSC dopaminergic neurons do not express TLR4.**

Thank you for this comment. As many TLR family members, TLR4 is also expressed in neurons^{18, 19, 20}. Interestingly, TLR4 signaling seems to be very active in DA neurons. We have confirmed TLR4 expression in iPSC-derived dopaminergic neurons using immunofluorescence (Figure 3).

Rabbit Anti TLR4 (20HCLC) ThermoFisher Scientific (cat. 710185), 2 μ g/ml

Figure 3. TLR4 expression in human iPSC-derived neurons.

- **Q3. Fig 2B and C: In Fig 2B, NFAT luciferase activity is greatly reduced in cells expressing LRRK2 or LRRK2-G2019S in the presence of PMA and ionomycin compared to control condition, however in Fig 2C these values are brought to 1 and reduction by IFN- γ ascertain what is the actual reduction in NFAT activity by IFN- γ in the presence of LRRK2.**

Thank you for pointing this out, we agree that data presentation was not optimal in previously submitted Fig. 2. However, in the revised version of our manuscript this experimental part has been replaced by data showing NFAT translocation using nuclear fractionation followed by Western blot analysis. To make actual differences immediately clear, all the data are now represented without further normalization.

- **Q3. In Fig. 2D images showing NFAT staining in nucleus are not convincing - staining appears cytoplasmic or perinuclear – this compared to clear nuclear staining in LRRK2 Ko condition (Fig. 2G). Would PMA/ionomycin treatment enhance nuclear translocation? IFN γ treatment appears to be toxic to cells particularly in LRRK2-G2019S condition (Fig. 2D).**

We agree with the reviewer that immunofluorescence staining of endogenous levels would be more relevant. We have tested several commercial anti-NFAT3 antibodies; however, the lack of specific signal prevented us from employing them for such analysis. Due to this restriction, we initially employed overexpression of an NFAT-GFP reporter and immunofluorescent quantification in iPSC-derived neurons, being however aware of the limitations of the approach. Hence, along with the biochemical data initially provided in Supplementary Fig. 1, we have now performed additional experiments, quantifying the endogenous NFAT localization, upon nuclear and cytosolic fractionation. These biochemical data further support decreased NFAT nuclear shuttling in LRRK2 mutant iPSC-derived neurons and the negative impact of IFN- γ . Furthermore, in the revised version of the manuscript, we provide additional data related to the activation pathway of NFAT by stimulation with ionomycin. These additional data that are now provided in Fig. 2 and Supplementary Fig. 1 further strengthen the link between LRRK2 and NFAT activity. Finally, we also provide data relative to the levels of phospho-NFAT3, which is the cytosolic, inactive form. Increased levels of phospho-NFAT3 were found in LRRK2 G2019S neurons. These additional data, which are now provided Supplementary Fig. 3, further strengthen the link between LRRK2 and NFAT activity.

- **Q4. Fig 3: Baseline intracellular Ca²⁺ levels in LRRK2-G2019S neurons are high suggesting Ca²⁺ buffering impairment and the Ca²⁺ influx in LRRK2-G2019S neurons following KCl treatment is lower – these results are opposite to those reported by Korecka et al 2018 – is there an explanation?**

We respectfully disagree with the reviewer's point of view in this comment. Korecka et al. (2018) reports ER Ca²⁺-storage capacity impairment in LRRK2-G2019S neurons, which is in line with our results. However, the reported higher Ca²⁺ release upon KCl in LRRK2-G2019S neurons is not directly comparable to our data, due to a different experimental design. In fact, while in Korecka et al. the authors first deplete the ER with long-term thapsigargin treatment (24 hrs) followed by KCl depolarization (two applications), our recordings have been performed upon addition of thapsigargin.

- **Q6. The authors have observed neurite shortening by LRRK2-G2019S and INF γ that is reversed by neuregulin – does neuregulin correct abnormal Ca²⁺ levels and ER buffering capacity in these cells? Are dopaminergic neurons particularly vulnerable to these effects of LRRK2-G2019S and INF- γ ?**

Thank you for this suggestion. We have now performed a new set of Ca²⁺ imaging experiments in dopaminergic neurons upon neuregulin treatment. Our results revealed that neuregulin had a significant rescue effect in ER storage capacity but did not affect the basal Ca²⁺ levels. These results are included in Supplementary Fig. 5. Concerning selective vulnerability, based on our data, we are not able to claim a selective vulnerability of DA neurons. This is the reason why we opted for a more general title "Interferon- γ signaling synergizes with LRRK2 in human neurons and microglia". However, the Reviewer is correct that an intriguing hypothesis would certainly be that DA neurons are selectively vulnerable to age- and inflammatory challenges based on their Ca²⁺ handling properties; we have included this point the revised discussion.

REFERENCES

1. Mogi M, Kondo T, Mizuno Y, Nagatsu T. p53 protein, interferon-gamma, and NF-kappaB levels are elevated in the parkinsonian brain. *Neuroscience letters* **414**, 94-97 (2007).
2. Ivanyuk D, et al. Loss of function of the mitochondrial peptidase PITRM1 induces proteotoxic stress and Alzheimer's disease-like pathology in human cerebral organoids. *bioRxiv*, accepted doi.org/10.1101/2020.01.27.919522, (2019).
3. Graef IA, et al. L-type calcium channels and GSK-3 regulate the activity of NF-ATc4 in hippocampal neurons. *Nature* **401**, 703-708 (1999).

4. Wild AR, Sinnen BL, Dittmer PJ, Kennedy MJ, Sather WA, Dell'Acqua ML. Synapse-to-Nucleus Communication through NFAT Is Mediated by L-type Ca²⁺ Channel Ca²⁺ Spike Propagation to the Soma. *Cell reports* **26**, 3537-3550 e3534 (2019).
5. Vaeth M, *et al.* Ca²⁺ Signaling but Not Store-Operated Ca²⁺ Entry Is Required for the Function of Macrophages and Dendritic Cells. *J Immunol* **195**, 1202-1217 (2015).
6. Graef IA, Chen F, Crabtree GR. NFAT signaling in vertebrate development. *Curr Opin Genet Dev* **11**, 505-512 (2001).
7. Bauer M, *et al.* Prevention of interferon-stimulated gene expression using microRNA-designed hairpins. *Gene Ther* **16**, 142-147 (2009).
8. Carrion MDP, *et al.* The LRRK2 G2385R variant is a partial loss-of-function mutation that affects synaptic vesicle trafficking through altered protein interactions. *Sci Rep* **7**, 5377 (2017).
9. Li Z, *et al.* IQGAP1 promotes neurite outgrowth in a phosphorylation-dependent manner. *The Journal of biological chemistry* **280**, 13871-13878 (2005).
10. Swiech L, *et al.* CLIP-170 and IQGAP1 cooperatively regulate dendrite morphology. *J Neurosci* **31**, 4555-4568 (2011).
11. Lin CH, Tsai PI, Wu RM, Chien CT. LRRK2 G2019S mutation induces dendrite degeneration through mislocalization and phosphorylation of tau by recruiting autoactivated GSK3 α . *J Neurosci* **30**, 13138-13149 (2010).
12. Kawakami F, *et al.* Leucine-rich repeat kinase 2 regulates tau phosphorylation through direct activation of glycogen synthase kinase-3 β . *FEBS J* **281**, 3-13 (2014).
13. Ohta E, *et al.* I2020T mutant LRRK2 iPSC-derived neurons in the Sagamihara family exhibit increased Tau phosphorylation through the AKT/GSK-3 β signaling pathway. *Human molecular genetics* **24**, 4879-4900 (2015).
14. Lin CH, *et al.* Lovastatin protects neurite degeneration in LRRK2-G2019S parkinsonism through activating the Akt/Nrf pathway and inhibiting GSK3 β activity. *Human molecular genetics* **25**, 1965-1978 (2016).
15. Mauro C, *et al.* NF- κ B controls energy homeostasis and metabolic adaptation by upregulating mitochondrial respiration. *Nat Cell Biol* **13**, 1272-1279 (2011).
16. Birdsall HH. Induction of ICAM-1 on human neural cells and mechanisms of neutrophil-mediated injury. *The American journal of pathology* **139**, 1341-1350 (1991).
17. Hery C, Sebire G, Peudener S, Tardieu M. Adhesion to human neurons and astrocytes of monocytes: the role of interaction of CR3 and ICAM-1 and modulation by cytokines. *Journal of neuroimmunology* **57**, 101-109 (1995).
18. Aurelian L, Warnock KT, Balan I, Puche A, June H. TLR4 signaling in VTA dopaminergic neurons regulates impulsivity through tyrosine hydroxylase modulation. *Transl Psychiatry* **6**, e815 (2016).

19. Calvo-Rodriguez M, de la Fuente C, Garcia-Durillo M, Garcia-Rodriguez C, Villalobos C, Nunez L. Aging and amyloid beta oligomers enhance TLR4 expression, LPS-induced Ca²⁺ responses, and neuron cell death in cultured rat hippocampal neurons. *Journal of neuroinflammation* **14**, 24 (2017).
20. Korgaonkar AA, *et al.* Toll-like Receptor 4 Signaling in Neurons Enhances Calcium-Permeable alpha-Amino-3-Hydroxy-5-Methyl-4-Isoxazolepropionic Acid Receptor Currents and Drives Post-Traumatic Epileptogenesis. *Annals of neurology* **87**, 497-515 (2020).

REVIEWER COMMENTS

Reviewer #1 (Remarks to the Author):

The revised manuscript improved and is now based on data acquired in disease-relevant cell types, namely iPSC-derived neurons and microglia, which profoundly increases the relevance of the findings. The authors performed an extensive amount of new experiments and answered the majority of my concerns appropriately. Particularly, it is worth to mention the fairness of the authors in sharing the negative results, or reporting technical limitations of IQGAP1 knockdown experiments in iPSC neurons. The current manuscript now describes different mechanisms affecting neurons and microglia carrying a pathogenic variant in LRRK2. Specifically a mechanism is described by which Interferon- γ has an impact on the phosphorylation of Akt3 and the shuttling of NFAT into the nucleus, this in turn affecting the neurite complexity of iPSC derived neurons and the cytokineproduction in microglia. The authors attribute the defects in NFAT shuttling partly to an altered Ca²⁺ homeostasis, and to altered microtubule network. In the manuscript the authors present several interesting observations, showing how these pathogenic variants in LRRK2 affect different pathways. Although the individual findings are quite interesting, the link between the mechanisms in the two different cell types is not clear enough and a big picture is difficult to see. Moreover, with the inclusion of additional experiments, the flow of the manuscript is even harder to follow.

1) The authors put a lot of effort in showing the dysregulation of Akt3. However, the link between this piece of data and the rest of the manuscript is hard to follow. This link needs clarification. 2) Especially showing upregulated mRNA levels of Akt3, unchanged levels of Akt3 in Western blot, and dysregulated levels of phospho-Akt3 by Luminex assay need further explanation. Finally in the supporting figures the authors show dysregulated phosphorylated-Akt instead of Akt3. This part of data would be clearer by adding the phospho-Akt3 by western blot. Moreover, concerning Fig. 1G and Suppl. Fig. 1G-H: In order to make a conclusion about differential effect of IFN γ on AKT3 and AKT phosphorylation in LRRK2 G2019S compared to control neurons, the authors need to provide a delta values of IFN γ -mediated effects, which would represent fold changes of phospho-AKT3 or phospho-AKT levels in IFN γ -treated vs. untreated neurons) in addition. The presentation of the data in the current form allows to see that phospho-AKT3 (phospho-AKT) levels are the lowest in LRRK2 G2019S neurons treated with IFN γ .

3) The findings regarding Ca²⁺ homeostasis are interesting, however not clear. The authors show that the Ca²⁺ baseline is shifted, having G2019S neurons a higher baseline. The authors use KCl to depolarize the neurons and stimulate the entrance of Ca²⁺. The authors claim that the Ca²⁺ influx is reduced with a longer recovery time. However, the timespan that the authors use to measure the influx confuses the results. Had the authors measured the total influx until recovery, how would this change the results? the total Ca²⁺ influx might probably be higher and not lower? Also the use of KCl affects rather neuronal excitability and not the ER-specific Ca²⁺ homeostasis. This raises the question whether the difference in Ca²⁺ influx are caused by differences in excitability (e.g. neurons being longer depolarized) rather than ER-buffering as claimed. The second set of measurements using thapsigargin to analyze the amount of Ca²⁺ influx is more accurate and much more convincing. I would suggest to put more focus on the second part of the data (thapsigargin) and rather exclude the KCl data since these measurements are unspecific to ER and open more questions than answer ones. It is important here to show the traces, and provide more information about buffers used during the experiment (extracellular and intracellular media composition).

4) Since IFN- γ has no influence on the depletion status of the ER in G2019S and the claim is that NFAT translocation is Ca²⁺ based, it was surprising that IFN- γ has an effect on neurite outgrowth in G2019S cells. This finding needs further explanation.

5. Please provide the clearer position indication of the protein marker values in the Western blot pictures (Fig.1 and Fig. 2) by placing a horizontal line between the kDa number and blot picture (as it was done in Fig. 1B but unfortunately only for the 250 kDa).

6. There are still some inconsistencies with the NFAT3 results in cellular subfractions of neurons

(presented in Fig. 2): - in the Rebuttal letter the authors explain that there is "a lack of validated anti-human NFAT3 antibodies", which was the reason to re-think immunofluorescence NFAT3 analyses in iPSC neurons and to concentrate on the biochemical NFAT3 detection in nuclear vs. cytoplasmic neuronal fractions by Western blot. Q.1: How the specificity/validation of the NFAT3 signals acquired by Western blot can be proven in the situation of the lack of validated NFAT3 antibodies? Are the anti-NFAT3 antibodies validated for Western blot but not immunofluorescence staining? Q.2: The answer of the authors to my comment of the differences of NFAT3 signals in the whole cell lysates and in the nuclear and cytoplasmic fractions (still present in Fig. 2 between 2A,B and 2C,D) was not clearly and properly answered. It is still unclear to me why NFAT3 appears as a collection of several bands in cytoplasmic and nuclear fractions (Fig. 2A,B) and as one band in the whole cell lysates (Fig. 2C,D)? Q.3: Please clarify the quantifications of the Western blot signals in Fig. 2 – NFAT or specifically NFAT3?

6. The time of latex beads incubation with microglia cells in Phagocytosis assay needs to be provided in the Mat&Methods part.

Minors:

Figure 1F. Mirror the westernblot to have the controls on the same side.

Figure 2A and B) The bands of the PAPRP1 western blot seem to be shifted or at a different magnification and not corresponding to the upper or lower lanes.

Reviewer #2 (Remarks to the Author):

The authors added a huge amount of data and strengthen the study. They have now connected more the observed phenotypes.

-However, their working model still remains confusing. It would be helpful to state clearly for instance their hypothesis regarding the temporal relationship of the observed phenotypes during PD progression. Along those lines, what are the new therapeutic interventions they propose based on their observations?

-Also it is not clear if the Ca⁺ and microtubule phenotypes are interconnected or two independent events? If two separate mechanisms what determines the one or the other will lead to the deregulation they observe?

-They claim that IFN- γ is the link between inflammation and neurodegeneration but the source of IFN- γ that leads to these phenotypes is not yet known.

-How these observations fit into the vulnerability of SNc neurons as they observe similar effects in cortical neurons?

- I would also like to see a proof of principle experiment of LRRK2 kinase inhibition using the specific dose of MLI-2 in these specific cells (Ser932 or pRabs could work). There are variable results across different cell lines.

Overall: a clear cut working model and its functional implications on PD based on their data and the literature will make the study stronger.

Reviewer #3 (Remarks to the Author):

The authors did a heroic effort responding to the reviewers' comments. This is an extremely important mechanistic paper on the role of LRRK2 in inflammation. I would like to see this published as soon as possible because it will be very important for the field.

My only comments are that I don't like the focus on the neurite shortening in the abstract. The data on the LRRK2 signaling pathways is much more important. The data in the paper on neurite length is not particularly robust. It may be that mutant LRRK2 influences synapse formation/pruning, but this cannot be analyzed with differentiated iPSCs that are a neurodevelopmental model and don't form

mature synapses. Also, Dr. Benson's group demonstrated after several days in cultures, neurons from G2019S-LRRK2 do not show differences in length. (PMID: 23646112).

Also, in the abstract, the authors focus on DA neurons but as they point out, they also examine glutamatergic neurons. LRRK2 is highly expressed in glutamatergic neurons (PMID: 24633735), mutant LRRK2 mostly affects the physiology of these neurons (PMID: 30249796)(PMID: 28930069) and frankly, I wish my colleagues would focus on the circuitry involved in PD and not just specifically on DA neurons.

I would just like to see the abstract and some of the text re-written to de-emphasize the neurite length data and the focus on DA neurons only.

Reviewer #4 (Remarks to the Author):

The new data in this revised manuscript, particularly on cytoplasmic/nuclear NFAT shuttling as presented in Figure 2, microtubule dynamics in iPSC-derived neurons in Figure 4, rescue experiment with NRG1 on neurite length in Figure 5 as well as NFAT-independent actions of LRRK2 and LRRK2G2019S in microglia, has considerably improved the manuscript. The revised presentation of the Results and Discussion sections has made the novelty of the work clearer and easier to understand.

"Interferon- γ signaling synergizes with LRRK2 in human neurons and microglia "

#Reviewer 1

The current manuscript now describes different mechanisms affecting neurons and microglia carrying a pathogenic variant in LRRK2. Specifically a mechanism is described by which Interferon- γ has an impact on the phosphorylation of Akt3 and the shuttling of NFAT into the nucleus, this in turn affecting the neurite complexity of iPSC derived neurons and the cytokine production in microglia. The authors attribute the defects in NFAT shuttling partly to an altered Ca²⁺ homeostasis, and to altered microtubule network. In the manuscript the authors present several interesting observations, showing how these pathogenic variants in LRRK2 affect different pathways. Although the individual findings are quite interesting, the link between the mechanisms in the two different cell types is not clear enough and a big picture is difficult to see. Moreover, with the inclusion of additional experiments, the flow of the manuscript is even harder to follow.

We thank the Reviewer for the overall positive comments. We also acknowledge that a cut-working model was missing in our previous version. We have addressed all these points, including a more detailed description of the link of IFN- γ -related mechanisms in different cell types. We describe the potential therapeutic implications of our findings and we highlight open questions. As suggested by Reviewer 2, we also provide possible explanations for the susceptibility of dopaminergic neurons to these mechanisms.

"Given the presence of T-lymphocytes in PD brains and the growing evidence pointing towards a role for T-cell responses in disease pathogenesis^{1, 2}, T-cells could be the main source of IFN- γ , which, in turn, induces LRRK2 expression in neurons and microglia, leading to neuronal damage and inflammatory reactions. Interestingly, a recent paper reported high α -syn-specific T-cell responses in PD patients at preclinical stage³. This early T-cell activation may be associated with aging and additional inflammatory events encountered over a lifetime (i.e., infections). Since reactive T-cells produce significant amounts of IFN- γ , these cells might serve as the primal source of IFN- γ production, driving immune responses, including the metabolic reprogramming and activation of microglia, as well as neuronal dysfunction, which is partially mediated by NFAT-related mechanisms. Thus, further studies are needed to dissect the signature of distinct subpopulations of T-cells responsible for early immune activation in PD. As indicated by our data and data from the literature, LRRK2 also modulates other essential microglial functions, including motility and phagocytosis. Interestingly, MHC-I is highly expressed

by substantia nigra DA neurons and locus coeruleus noradrenergic neurons⁴. Furthermore, primary DA murine neurons and human embryonic stem cells (hESCs) are more susceptible to MHC-I induction by IFN- γ than other neuronal populations⁴. These mechanisms would render catecholaminergic neurons selective targets for immune-mediated cell death, as observed in viral parkinsonism⁵. In this scenario, LRRK2 G2019S individuals may be even more vulnerable to IFN- γ -driven mechanisms."

1) The authors put a lot of effort in showing the dysregulation of Akt3. However, the link between this piece of data and the rest of the manuscript is hard to follow. This link needs clarification.

2) Especially showing upregulated mRNA levels of Akt3, unchanged levels of Akt3 in Western blot, and dysregulated levels of phospho-Akt3 by Luminex assay need further explanation.

Thank you for this comment. Concerning the role of AKT(3), we have now further clarified its role and link to NFAT3 modulation. AKT family members play a key role in a variety of cellular processes. As concerns our study, AKT phosphorylates and inactivates GSK3 β , one of the kinases responsible for NFAT phosphorylation, leading to its cytosolic shuttling and inactivation. Our data show that IFN- γ leads to a decrease in AKT(3) phosphorylation/activation and a decrease in NFAT3 shuttling. It is important to note that IFN- γ significantly reduced the phosphorylation of total AKT. Hence, our data suggest that IFN- γ negatively regulates the nuclear shuttling of NFAT3, and this happens, at least in part, through the AKT/GSK3 β pathway. Interestingly, it has also been shown that LRRK2 kinase activity is associated with GSK3 β activation in a variety of model systems, including human iPSC-derived neurons^{6, 7, 8, 9}. The Reviewer is correct: we also expected a correspondence between AKT3 RNA and protein levels. However, it is likely that the increase of AKT3 expression is a feedback mechanism, linked to its decreased phosphorylation state, hence reduced activity. Concerning the decrease of phospho-AKT(3) upon IFN- γ treatment, this corroborates previous findings showing that IFN- γ suppresses AKT signaling in human macrophages¹⁰. We have now modified the manuscript to make all these points more clear.

Q3) Finally in the supporting figures the authors show dysregulated phosphorylated-Akt instead of Akt3. This part of data would be clearer by adding the phospho-Akt3 by western blot.

Unfortunately, there is no validated phospho-Akt3 specific antibody for Western blotting. Thus, we used the Milliplex Magnetic Bead Panel to specifically quantify the protein levels of phospho-

Akt3 and total-Akt3. Additionally, we evaluated the levels of phospho-Akt1/2/3 using Western blotting.

Q4) Moreover, concerning Fig. 1G and Suppl. Fig. 1G-H: In order to make a conclusion about differential effect of IFN γ on AKT3 and AKT phosphorylation in LRRK2 G2019S compared to control neurons, the authors need to provide a delta values of IFN γ -mediated effects, which would represent fold changes of phospho-AKT3 or phospho-AKT levels in IFN γ -treated vs. untreated neurons) in addition. The presentation of the data in the current form allows to see that phospho-AKT3 (phospho-AKT) levels are the lowest in LRRK2 G2019S neurons treated with IFN γ .

Thank you for this suggestion. We have expressed the values as fold changes over untreated control, in order to provide information about differences in basal levels of AKT(3) in LRRK2 G2019S compared to control neurons. However, we do agree that it would be useful to provide information on the relative reduction of AKT phosphorylation in LRRK2 G2019S and control neurons. Raw data are provided in the source data file; for clarity, we have now included this information as a percentage in the text, as follows: "IFN- γ treatment significantly reduced AKT3 phosphorylation in both LRRK2 G2019S and control neurons, with the strongest effect observed in LRRK2 G2019S neurons (38.9% reduction, compared to 28.8% in control neurons) (Fig. 1G)" and "... with a more pronounced effect in LRRK2 G2019S neurons (33.2% reduction, compared to 23.7% in control neurons) (Supplementary Fig. 1G, H)".

5) The findings regarding Ca $^{2+}$ homeostasis are interesting, however not clear. The authors show that the Ca $^{2+}$ baseline is shifted, having G2019S neurons a higher baseline. The authors use KCl to depolarize the neurons and stimulate the entrance of Ca $^{2+}$. The authors claim that the Ca $^{2+}$ influx is reduced with a longer recovery time. However, the timespan that the authors use to measure the influx confuses the results. Had the authors measured the total influx until recovery, how would this change the results? The total Ca $^{2+}$ influx might probably be higher and not lower?

We thank the reviewer for this comment. Regarding the recording time, the total influx -until complete recovery- was measured. More specifically, the total recording time following KCl application was 15 minutes. However, Ca $^{2+}$ levels, after reaching the peak, were restored to baseline within a timespan shorter than 10 minutes. Thus, we would not expect any difference in our results upon longer recording periods. Concerning the total Ca $^{2+}$ influx, the reviewer is correct: Δ -Ca $^{2+}$ was measured as the relative amplitude of the induced Ca $^{2+}$ peak (peak cytosolic calcium minus the corresponding baseline, measured as a change in Fura-2 fluorescence) and

not as the total calcium influx. Please note that, as per Reviewer's suggestions, we have now removed KCl data.

6) Also the use of KCl affects rather neuronal excitability and not the ER-specific Ca²⁺ homeostasis. This raises the question whether the difference in Ca²⁺ influx are caused by differences in excitability (e.g. neurons being longer depolarized) rather than ER-buffering as claimed. The second set of measurements using thapsigargin to analyze the amount of Ca²⁺ influx is more accurate and much more convincing. I would suggest to put more focus on the second part of the data (thapsigargin) and rather exclude the KCl data since these measurements are unspecific to ER and open more questions than answer ones.

The Reviewer is correct, KCl-dependent Ca²⁺ dynamics, as evaluated in our study, address neuronal excitability rather than intracellular Ca²⁺ buffering mechanisms. For this reason, we have removed these data. As suggested, we have now focused more on the results concerning ER-Ca²⁺ stores, as assessed upon thapsigargin application.

7) It is important here to show the traces, and provide more information about buffers used during the experiment (extracellular and intracellular media composition).

We would like to thank the reviewer for the constructive suggestion concerning our data display. We have now included the traces (Fig.3A and C, Supplementary Fig. 5C). Additional details on the recording medium composition have now been added to the Materials & Methods section. We would like to point out that the measurements performed in our studies are live-cell calcium imaging, hence no "intracellular recording medium" was employed.

8) Since IFN- γ has no influence on the depletion status of the ER in G2019S and the claim is that NFAT translocation is Ca²⁺ based, it was surprising that IFN- γ has an effect on neurite outgrowth in G2019S cells. This finding needs further explanation.

NFAT nuclear translocation is a dynamic process depending on intracellular Ca²⁺ fluctuations¹¹. One of the key pathways modulating NFAT3 is calcium mobilization involving the activation of store-operated Ca²⁺ entry (SOCE)¹², which operates upstream of both calcineurin and NFAT3. Hence, acute ER depletion promotes NFAT3 translocation. Here, we show that that *LRRK2* G2019S leads to defects in ER Ca²⁺ storage in neurons, attenuating Ca²⁺ release from the ER in response to TPH. In general, G2019S neurons are less sensitive to any stimulus that leads to release of Ca²⁺ from the ER stores. Interestingly, Korecka et al. found a significant decrease in expression levels of the ER Ca²⁺ sensor STIM1, which links ER Ca²⁺ store depletion and SOCE

activation in G2019S neurons¹³. The authors also show a reduced functionality of SOCE as a possible consequence of the decreased STIM1 expression and provide a link between ER Ca²⁺ levels and neurite defects¹³. Hence, our findings, along with data from the literature, point towards a role of Ca²⁺ dynamics in defects of NFAT3 translocation in G2019S neurons. In addition, we examined the impact of chronic treatment with IFN- γ on ER Ca²⁺ release and we found that IFN- γ treatment leads to a depletion of intracellular Ca²⁺ ER stores. Since the ER was already depleted at baseline, we did not find significant differences upon IFN- γ in G2019S neurons. These data further support the hypothesis that G2019S mutation leads to defects in intracellular calcium ER stores, hence NFAT3 nuclear translocation. In addition, these findings suggest a possible mechanism by which a chronic inflammatory environment, as mimicked in our study by 24 hours IFN- γ treatment, could lead to defects in NFAT3 shuttling. While we hypothesize that the impact of *LRRK2* G2019S on NFAT3 shuttling can be, at least in part, mediated by defects in the ER Ca²⁺ stores, based on these data, we do not claim that the differential impact of IFN- γ on G2019S neurons is mediated by Ca²⁺-related mechanisms. However, NRG1 treatment had a positive impact on the recovery of the ER Ca²⁺ stores in G2019S neurons and rescues neurite shortening caused by IFN- γ , supporting the initial hypothesis that defects in ER Ca²⁺ homeostasis can contribute to a reduction in NFAT3 shuttling in G2019S neurons. Finally, we do agree that there are multiple mechanisms by which IFN- γ modulates NFAT3 shuttling, including proteasome degradation and reduced AKT(3) signaling. We have now made these points more clear in the manuscript.

9) Please provide the clearer position indication of the protein marker values in the Western blot pictures (Fig.1 and Fig. 2) by placing a horizontal line between the kDa number and blot picture (as it was done in Fig. 1B but unfortunately only for the 250 kDa).

Thank you for pointing this out. Appropriate modifications have been made to the revised Figures.

10) There are still some inconsistencies with the NFAT3 results in cellular subfractions of neurons (presented in Fig. 2): - in the Rebuttal letter the authors explain that there is “a lack of validated anti-human NFAT3 antibodies”, which was the reason to re-think immunofluorescence NFAT3 analyses in iPSC neurons and to concentrate on the biochemical NFAT3 detection in nuclear vs. cytoplasmic neuronal fractions by Western blot.

11) How the specificity/validation of the NFAT3 signals acquired by Western blot can be proven in the situation of the lack of validated NFAT3 antibodies? Are the anti-NFAT3 antibodies validated for Western blot but not immunofluorescence staining?

We would like to apologize that this point was not clear in the previous letter. We have tested several antibodies against human NFAT3. While these antibodies have been successfully utilized in Western blotting, after careful assessment we found them to react in a rather unspecific manner in immunocytochemistry. The antibody used in this study (NFAT3 (23E6), Rabbit, #2183, Cell Signaling) is validated for Western blotting analysis and specific for NFAT3. Indeed, the Western blotting bands presented in Figure 2 and Supplementary Figure 1 are specific and at the expected molecular weight (100-140 kDa). However, this antibody is not suitable for immunofluorescence staining.

12) The answer of the authors to my comment of the differences of NFAT3 signals in the whole cell lysates and in the nuclear and cytoplasmic fractions (still present in Fig. 2 between 2A, B and 2C, D) was not clearly and properly answered. It is still unclear to me why NFAT3 appears as a collection of several bands in cytoplasmic and nuclear fractions (Fig. 2A, B) and as one band in the whole cell lysates (Fig. 2C, D)?

Thank you for pointing this out, since our previous answer was not sufficient. As previously mentioned, the Reviewer's comment that the signal is different is correct. It is not uncommon that different band patterns are identified in cellular fractionation compared to the whole cell lysate. In general, cellular fractionation allows a better separation and identification of phosphorylated vs de-phosphorylated forms and different protein isoforms. The difference in band pattern is mostly due to technical differences in the procedure as summarized in the table below:

	Whole cell lysates	Fractionated samples
Buffers used	NP-40	NE/PER Nuclear & cytosolic extraction kit
Gel used	7.5%	4-12% gradient
Running time	60 min	120 min
Protein amount	30µg	10µg

The obtained signals comply with the ones reported in the literature and technical datasheet of the antibody (please see below).

<https://www.cellsignal.com/product/productDetail.jsp?productId=2183>

13) Please clarify the quantifications of the Western blot signals in Fig. 2 – NFAT or specifically NFAT3?

We would like to apologize for this inexplicit point. It has now been changed to “nuclear/total NFAT3” on the graph axis.

14) The time of latex beads incubation with microglia cells in Phagocytosis assay needs to be provided in the Mat&Methods part.

Thank you. This information has now been added to the Materials & Methods section.

Minors:

Figure 1F. Mirror the western blot to have the controls on the same side.

Thank you, this correction has been done.

Figure 2A and B) the bands of the PAPRP1 western blot seem to be shifted or at a different magnification and not corresponding to the upper or lower lanes.

Thank you, this correction has been done.

Reviewer #2 (Remarks to the Author):

The authors added a huge amount of data and strengthen the study. They have now connected more the observed phenotypes.

Q1. However, their working model still remains confusing. It would be helpful to state clearly for instance their hypothesis regarding the temporal relationship of the observed phenotypes during PD progression. Along those lines, what are the new therapeutic interventions they propose based on their observations?

Q2. They claim that IFN- γ is the link between inflammation and neurodegeneration but the source of IFN- γ that leads to these phenotypes is not yet known.

We thank the Reviewer for this suggestion, and we have now addressed these issues. Based on our data and recent data from the literature, we propose that immune mechanisms may be an early event in PD pathogenesis. In our study we have not observed increased production of IFN- γ by activated LRRK2 G2019S compared to microglia. Since our study has been conducted *in vitro*, we cannot exclude that aged LRRK2 G2019S microglia respond differently to inflammatory stimuli. These studies are currently ongoing in our lab. However, it is also likely that other immune cell types may be the primary source of IFN- γ . The most important producers of IFN- γ are T lymphocytes and NK cells. In particular, T-cells might be the population playing an instrumental role by polarizing microglia toward a pathogenic phenotype via secretion of IFN- γ . Interestingly, a very recent paper has reported high α -syn-specific T-cell responses in PD patients at the preclinical stage³. This early T activation may be associated with ageing, as well as additional inflammatory events encountered by the individual over a lifetime (i.e.- infections). Since reactive T-cells produce significant amounts of IFN- γ , T-cells may be the early source of IFN- γ and drive immune responses, including microglia metabolic reprogramming and activation as well as changes in motility, and neuronal dysfunction, which is partially mediated by NFAT-related mechanisms. However, as indicated by our data and data from the literature, LRRK2 also directly modulates microglia function at different levels, including motility and phagocytosis. From a therapeutic perspective, our findings indicate that modulating microglia metabolism (using for instance small molecules or metabolites to modulate glycolysis) might be an efficient strategy to regulate the functional phenotype of microglia prevailing in LRRK2 PD. Furthermore, immunotherapy approaches targeting T-cells may prove efficient in dampening this inflammatory cascade. Future studies, including single cell approaches, shall further characterize the immune cell subsets, including microglia and T-cells, involved in the early inflammatory events. Dissecting the specific immune cell subsets will help design novel strategies to modulate microglia pathogenic polarization and T cell activation.

Q3. Also it is not clear if the Ca⁺ and microtubule phenotypes are interconnected or two independent events? If two separate mechanisms what determines the one or the other will lead to the deregulation they observe?

We thank the Reviewer for this comment. It is well established that calcium interacts with and regulates the cytoskeleton, including microtubule dynamics at different levels^{14, 15, 16, 17}. For instance, calcium signaling in neuronal growth cones is essential for axon guidance^{18, 19, 20}. Vice versa, microtubule dynamics affect, directly or indirectly, calcium handling^{21, 22}. Interestingly, several works now link microtubules to SOCE. Specifically, microtubules have been shown to promote SOCE signaling pathway by optimizing the localization of STIM1²³. Given the crucial role of SOCE in NFAT shuttling, we hypothesize that in our model, LRRK2-related changes in microtubules affect calcium handling. We have now addressed this point in the discussion.

Q4. How these observations fit into the vulnerability of SNc neurons as the observed similar effects in cortical neurons?

Our findings may link IFN- γ related immune pathways and the selective vulnerability of midbrain neurons at multiple levels. MHC-I, whose expression is induced by IFN- γ , is highly expressed in substantia nigra DA neurons and locus coeruleus noradrenergic neurons⁴. Furthermore, primary DA murine neurons and DA neurons generated from human embryonic stem cells are more susceptible to IFN- γ -mediated MHC-I induction than other neuronal populations⁴. Hence, these mechanisms would render catecholaminergic neurons selective targets for immune mediated cell death, as observed in viral parkinsonism⁵. Furthermore, based on our findings, LRRK2 G2019S individuals may be even more vulnerable to IFN- γ driven mechanisms.

Q5. I would also like to see a proof of principle experiment of LRRK2 kinase inhibition using the specific dose of Mli-2 in these specific cells (Ser932 or pRabs could work). There are variable results across different cell lines.

Thank you for the suggestion. We have performed Western blotting analysis showing LRRK2 phospho-Ser935 levels in iPSC-derived neurons and microglia, upon Mli-2 treatment. Mli-2 treatment reduced the levels of LRRK2 S935 phosphorylation.

IPSC-derived microglia and neurons were treated with Mli-2 (100nM, 24hrs) and levels of LRRK2 (phospho S935) and total LRRK2 were assessed by Western blotting. The following antibodies were used: rabbit anti-LRRK2 (phospho S935) [UDD2 10(12)] (1:500, Abcam #ab133450); rabbit anti-LRRK2 [MJFF2 (c41-2)] (1:1000, Abcam, #ab133474); mouse anti-β-Actin (1:10000, Santa Cruz, #sc-47778).

Reviewer #3 (Remarks to the Author):

The authors did a heroic effort responding to the reviewers' comments. This is an extremely important mechanistic paper on the role of LRRK2 in inflammation. I would like to see this published as soon as possible because it will be very important for the field.

We thank the Reviewer for the positive comments and the interest in our work.

Q1. My only comments are that I don't like the focus on the neurite shortening in the abstract. The data on the LRRK2 signaling pathways is much more important. The data in the paper on neurite length is not particularly robust. It may be that mutant LRRK2 influences synapse formation/pruning, but this cannot be analyzed with differentiated iPSCs that are a neurodevelopmental model and don't form mature synapses. Also, Dr. Benson's group demonstrated after several days in cultures, neurons from G2019S-LRRK2 do not show differences in length. (PMID: 23646112).

We thank the Reviewer for these comments. We do agree that the impact of LRRK2 G2019S in our model system may affect additional aspects of neuronal homeostasis, including synapse formation. Concerning the cited paper, while no significant differences were observed in branching and motility, the authors were still able to observe defects in dendrite length at DIV 14 in primary neurons from LRRK2 G2019S mice. However, as correctly pointed out by Reviewer, the effect on dendrite length was modest, in line with our results and data from the literature. We agree with the Reviewer and we have now changed the focus and also highlight the role of LRRK2 at the synapse, which extends beyond the DA system.

Page 14, "Furthermore, it is important to note that both NFAT and LRRK2 are involved in synaptic plasticity, also beyond the DA system"^{24, 25}

Q3. Also, in the abstract, the authors focus on DA neurons but as they point out, they also examine glutamatergic neurons. LRRK2 is highly expressed in glutamatergic neurons (PMID: 24633735), mutant LRRK2 mostly affects the physiology of these neurons (PMID: 30249796) (PMID: 28930069) and frankly, I wish my colleagues would focus on the circuitry involved in PD and not just specifically on DA neurons.

Q4. I would just like to see the abstract and some of the text re-written to de-emphasize the neurite length data and the focus on DA neurons only.

Thank you for this comment. In the revised manuscript we have rephrased both the abstract and the main text to de-emphasize the DAcentric view and the focus on neurite length.

Reviewer #4 (Remarks to the Author):

The new data in this revised manuscript, particularly on cytoplasmic/nuclear NFAT shuttling as presented in Figure 2, microtubule dynamics in iPSC-derived neurons in Figure 4, rescue experiment with NRG1 on neurite length in Figure 5 as well as NFAT-independent actions of LRRK2 and LRRK2G2019S in microglia, has considerably improved the manuscript. The revised presentation of the Results and Discussion sections has made the novelty of the work clearer and easier to understand.

We would like to thank the Reviewer for taking the time to consider our manuscript and his valuable contribution during the revision process.

References

1. Brochard V, *et al.* Infiltration of CD4+ lymphocytes into the brain contributes to neurodegeneration in a mouse model of Parkinson disease. *J Clin Invest* **119**, 182-192 (2009).
2. Sommer A, *et al.* Th17 Lymphocytes Induce Neuronal Cell Death in a Human iPSC-Based Model of Parkinson's Disease. *Cell stem cell* **23**, 123-131 e126 (2018).
3. Lindestam Arlehamn CS, *et al.* alpha-Synuclein-specific T cell reactivity is associated with preclinical and early Parkinson's disease. *Nat Commun* **11**, 1875 (2020).
4. Cebrian C, *et al.* MHC-I expression renders catecholaminergic neurons susceptible to T-cell-mediated degeneration. *Nat Commun* **5**, 3633 (2014).
5. Deczkowska A, Baruch K, Schwartz M. Type I/II Interferon Balance in the Regulation of Brain Physiology and Pathology. *Trends in immunology* **37**, 181-192 (2016).
6. Lin CH, Tsai PI, Wu RM, Chien CT. LRRK2 G2019S mutation induces dendrite degeneration through mislocalization and phosphorylation of tau by recruiting autoactivated GSK3ss. *J Neurosci* **30**, 13138-13149 (2010).
7. Kawakami F, *et al.* Leucine-rich repeat kinase 2 regulates tau phosphorylation through direct activation of glycogen synthase kinase-3beta. *FEBS J* **281**, 3-13 (2014).
8. Ohta E, *et al.* I2020T mutant LRRK2 iPSC-derived neurons in the Sagamihara family exhibit increased Tau phosphorylation through the AKT/GSK-3beta signaling pathway. *Human molecular genetics* **24**, 4879-4900 (2015).

9. Lin CH, *et al.* Lovastatin protects neurite degeneration in LRRK2-G2019S parkinsonism through activating the Akt/Nrf pathway and inhibiting GSK3beta activity. *Human molecular genetics* **25**, 1965-1978 (2016).
10. Hu X, *et al.* IFN-gamma suppresses IL-10 production and synergizes with TLR2 by regulating GSK3 and CREB/AP-1 proteins. *Immunity* **24**, 563-574 (2006).
11. Sharma S, *et al.* An siRNA screen for NFAT activation identifies septins as coordinators of store-operated Ca²⁺ entry. *Nature* **499**, 238-242 (2013).
12. Vaeth M, *et al.* Store-Operated Ca(2+) Entry in Follicular T Cells Controls Humoral Immune Responses and Autoimmunity. *Immunity* **44**, 1350-1364 (2016).
13. Korecka JA, *et al.* Neurite Collapse and Altered ER Ca(2+) Control in Human Parkinson Disease Patient iPSC-Derived Neurons with LRRK2 G2019S Mutation. *Stem cell reports* **12**, 29-41 (2019).
14. Solomon F. Binding sites for calcium on tubulin. *Biochemistry* **16**, 358-363 (1977).
15. Dedman JR, Brinkley BR, Means AR. Regulation of microfilaments and microtubules by calcium and cyclic AMP. *Adv Cyclic Nucleotide Res* **11**, 131-174 (1979).
16. O'Brien ET, Salmon ED, Erickson HP. How calcium causes microtubule depolymerization. *Cell motility and the cytoskeleton* **36**, 125-135 (1997).
17. Gasperini RJ, *et al.* How does calcium interact with the cytoskeleton to regulate growth cone motility during axon pathfinding? *Molecular and cellular neurosciences* **84**, 29-35 (2017).
18. Gomez TM, Spitzer NC. In vivo regulation of axon extension and pathfinding by growth-cone calcium transients. *Nature* **397**, 350-355 (1999).
19. Zheng JQ. Turning of nerve growth cones induced by localized increases in intracellular calcium ions. *Nature* **403**, 89-93 (2000).
20. Wen Z, Guirland C, Ming GL, Zheng JQ. A CaMKII/calcineurin switch controls the direction of Ca(2+)-dependent growth cone guidance. *Neuron* **43**, 835-846 (2004).
21. Miragoli M, *et al.* Microtubule-Dependent Mitochondria Alignment Regulates Calcium Release in Response to Nanomechanical Stimulus in Heart Myocytes. *Cell reports* **14**, 140-151 (2016).

22. Khairallah RJ, *et al.* Microtubules underlie dysfunction in duchenne muscular dystrophy. *Science signaling* **5**, ra56 (2012).
23. Smyth JT, DeHaven WI, Bird GS, Putney JW, Jr. Role of the microtubule cytoskeleton in the function of the store-operated Ca²⁺ channel activator STIM1. *J Cell Sci* **120**, 3762-3771 (2007).
24. Benson DL, Matikainen-Ankney BA, Hussein A, Huntley GW. Functional and behavioral consequences of Parkinson's disease-associated LRRK2-G2019S mutation. *Biochem Soc Trans* **46**, 1697-1705 (2018).
25. Rajgor D, *et al.* Local miRNA-Dependent Translational Control of GABAAR Synthesis during Inhibitory Long-Term Potentiation. *Cell reports* **31**, 107785 (2020).

REVIEWERS' COMMENTS:

Reviewer #1 (Remarks to the Author):

My questions have been sufficiently addressed.

Reviewer #2 (Remarks to the Author):

The authors have made an amazing amount of new experiments which strengthen their manuscript since the first draft we saw. Given the significance of the conceptual focus and the technical improvements along the way, I believe this manuscript is now suitable for publication at Nature Communications.